# Molecular tuning of sea anemone stinging

Lily S He[1], Yujia Qi[2], Corey AH Allard[1], Wendy A Valencia-Montoya[1,3], Stephanie P Krueger[1], Keiko Weir[1], Agnese Seminara[2]*, Nicholas W Bellono[1]*

[1]Department of Molecular and Cellular Biology, Harvard University, Cambridge, United States; [2]Machine Learning Center Genoa (MalGa), Department of Civil, Chemical and Environmental Engineering (DICCA), University of Genoa, Genoa, Italy; [3]Department of Organismic and Evolutionary Biology and Museum of Comparative Zoology, Harvard University, Cambridge, United States

**Abstract** Jellyfish and sea anemones fire single-use, venom-covered barbs to immobilize prey or predators. We previously showed that the anemone *Nematostella vectensis* uses a specialized voltage-gated calcium (Ca$_V$) channel to trigger stinging in response to synergistic prey-derived chemicals and touch (Weir et al., 2020). Here, we use experiments and theory to find that stinging behavior is suited to distinct ecological niches. We find that the burrowing anemone *Nematostella* uses uniquely strong Ca$_V$ inactivation for precise control of predatory stinging. In contrast, the related anemone *Exaiptasia diaphana* inhabits exposed environments to support photosynthetic endosymbionts. Consistent with its niche, *Exaiptasia* indiscriminately stings for defense and expresses a Ca$_V$ splice variant that confers weak inactivation. Chimeric analyses reveal that Ca$_V\beta$ subunit adaptations regulate inactivation, suggesting an evolutionary tuning mechanism for stinging behavior. These findings demonstrate how functional specialization of ion channel structure contributes to distinct organismal behavior.

*For correspondence:
agnese.seminara@unige.it (AS);
nbellono@harvard.edu (NWB)

## eLife assessment

This is an **important** paper that links distinctive stinging behavior of two related anemones occupying different ecological niches to varying inactivation properties of voltage-gated calcium channels conferred by auxiliary Cavβ subunits. Further **convincing** evidence is provided that these differences are mediated by alternative splicing of Cavβ subunit of the calcium channel. The study will be of interest to scientists studying Ca2+ signaling, ion channel biophysicists, and marine biologists.

## Introduction

Sea anemones, jellyfish, corals, and hydrozoans of the Cnidarian phylum use specialized cells called nematocytes to sting for predation or defense. Mechanical and chemical stimuli from prey or predators act synergistically on nematocytes to mediate rapid discharge of a toxin-covered barb from its nematocyst organelle (*Holstein and Tardent, 1984*; *Watson and Mire-Thibodeaux, 1994*; *Babonis and Martindale, 2014*). Nematocyst discharge requires calcium (Ca$^{2+}$) influx and, as a one-time use organelle, is tightly controlled to prevent energetically wasteful stinging to irrelevant stimuli (*Lubbock et al., 1981*; *Gitter et al., 1994*; *Watson and Hessinger, 1994*). We previously found that the starlet sea anemone *Nematostella vectensis* uses a uniquely adapted voltage-gated Ca$^{2+}$ channel (Ca$_V$) to integrate simultaneously presented chemical and mechanical cues that elicit nematocyst discharge. *Nematostella* Ca$_V$ exhibits unusually 'strong' steady-state voltage-dependent inactivation at resting membrane voltages to reduce cellular excitability and prevent stinging behavior in response to extraneous, non-prey touch signals. Chimeric analyses of *Nematostella* and mammalian Ca$_V$ showed that the auxiliary β subunit (Ca$_V\beta$) is required and sufficient for low-voltage steady-state inactivation in

*Nematostella* Ca$_V$ channel complexes. Ca$_V$ inactivation is relieved by hyperpolarization of the nematocyte membrane potential to very negative voltages through the effect of prey-derived chemosensory signals that are synaptically transmitted from sensory neurons. Upon relieving Ca$_V$ inactivation, direct touch responses are amplified to trigger nematocyst discharge (*Weir et al., 2020*). Thus, single nematocytes integrate synergistic cues to elicit a precise response, representing a unique cellular system to study how cells detect and transduce signals to produce discrete behavior.

While Ca$_V$-mediated sensory integration represents one mechanism by which nematocytes 'decide' when to sting, the incredible diversity of cnidarian biology suggests that stinging behavior must be adapted to support the demands of different lifestyles. Cnidarian taxa occupy diverse environmental niches and endure specific metabolic demands, predatory challenges, and environmental pressures for survival, which results in distinct selective pressures on nematocyte evolution (*Beckmann and Özbek, 2012*; *Babonis et al., 2022*). *Nematostella vectensis* and *Exaiptasia diaphana* represent an example of closely related cnidarians with differing environmental niches and metabolic demands (*Darling et al., 2005*; *Bedgood et al., 2020*). *Nematostella* are found in shallow brackish water of coastal marshes where they are buried in the mud, hidden from predators, with only their tentacles exposed to catch unsuspecting passing prey (*Fraune et al., 2016*). Thus, we hypothesize that their stinging is under tight regulation adapted for opportunistic predation. In contrast, *Exaiptasia* are exposed to predators while living in shallow, open ocean environments that provide sufficient sunlight for their endosymbionts to produce important photosynthetic products and nutrients (*Baumgarten et al., 2015*). Considering these dramatically different ecological contexts, we hypothesized that Ca$_V$-mediated regulation of nematocyte discharge has adapted to reflect the demands on stinging behavior in these two anemones. We therefore probed the behavior of these related but distinct anemones and investigated how subtle tuning of a shared molecular-regulatory mechanism drives adaptation in physiology and behavior associated with niche diversification.

In this study, we find that the symbiotic anemone *Exaiptasia* stings in response to mechanical stimuli alone, independent of predation pressure. This behavior serves as a stark contrast with *Nematostella* stinging, which is only elicited by synergistic prey chemicals and touch. Markov decision process modeling coupled with behavioral experiments revealed that *Nematostella* stings as an optimal predator, whereas *Exaiptasia* exhibits optimal defensive stinging behavior. Consistent with indiscriminate stinging behavior, we discover that *Exaiptasia* nematocyte physiology lacks the unusual Ca$_V$ inactivation used by *Nematostella* to inactivate cells at rest and prevent responses to touch in the absence of prey chemicals. 'Weak' steady-state inactivation of *Exaiptasia* Ca$_V$ is mediated by a splice isoform of the beta subunit (Ca$_V\beta$) with a distinct N-terminus and allows for robust activation from resting membrane potentials. Analysis of chimeric jellyfish and anemone channels reveals that Ca$_V$ inactivation is broadly regulated by the Ca$_V\beta$ N-terminus, suggesting an evolutionary tuning mechanism that could contribute to specific stinging behavior across cnidarians. Thus, we propose Ca$_V$ adaptations as one molecular mechanism that could shift predatory versus defensive stinging in cnidarians. These results highlight how subtle adaptations in protein structure contribute to complex organismal behavior.

## Results

### Comparative sea anemone stinging behavior

In their natural habitat, *Nematostella* are hidden by burrowing within the sandy substrate and use an opportunistic predatory strategy to capture prey with their tentacles. In contrast, symbiotic *Exaiptasia* are found within open waters where they experience greater risk of predation and therefore must adopt a more defensive stance. Thus, we first asked whether differences in ecological pressure are reflected by stinging behavior. Consistent with our previous findings, we observed *Nematostella* stinging in response to simultaneously delivered prey extract and touch, reflecting stinging control adapted for predation (*Figure 1*; *Weir et al., 2020*). Strikingly, *Exaiptasia* tentacles instead exhibited robust stinging even in the absence of prey chemicals (touch alone, *Figure 1*). Similar touch-evoked stinging was observed for *Exaiptasia* acontia, which are defensive nematocyte-enriched structures that are ejected and release toxins to repel predators (*Lam et al., 2017*). Considering the drastic differences in stinging behavior, we wondered if *Exaiptasia*'s indiscriminate stinging reflects a distinct control strategy.

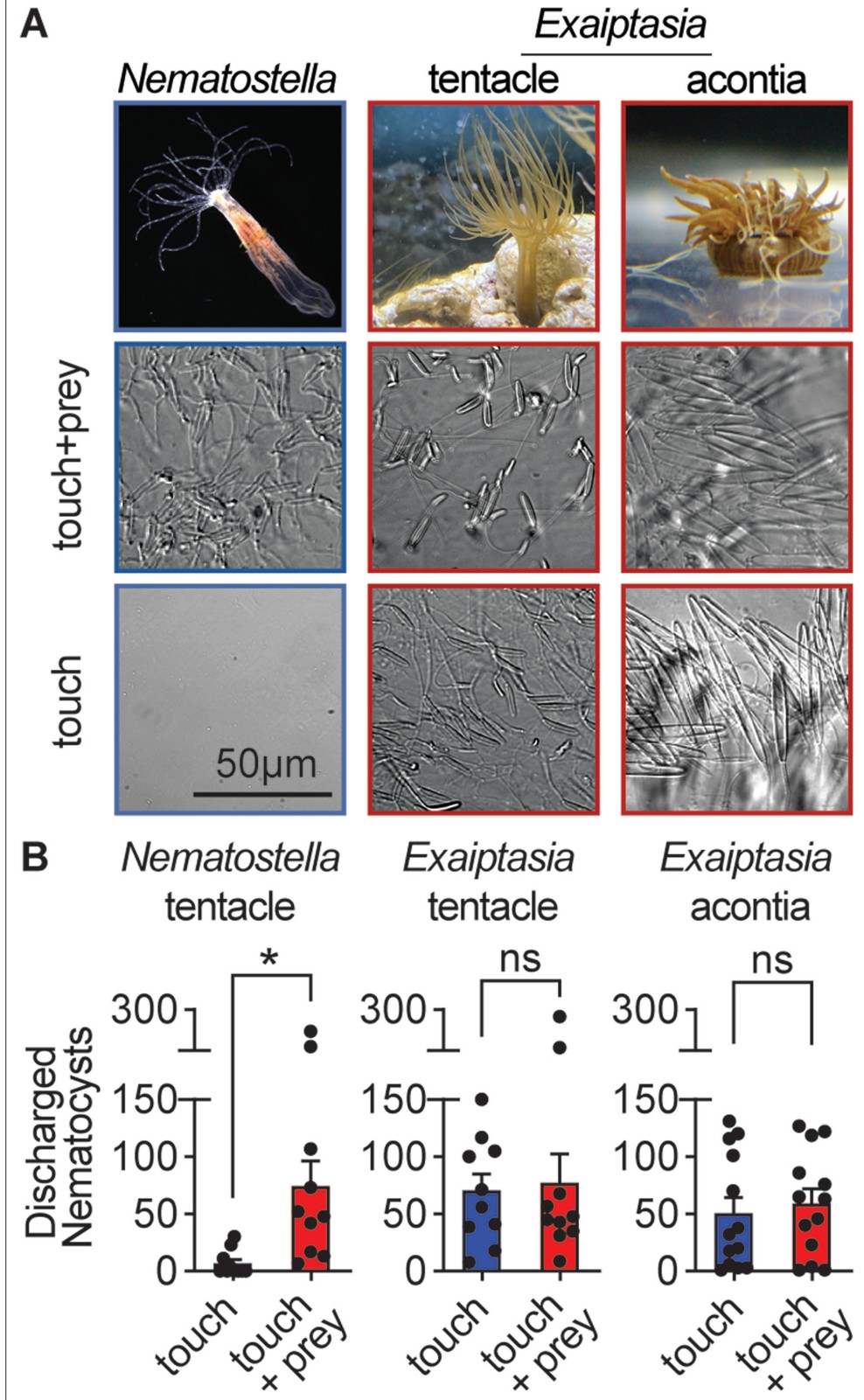

**Figure 1.** Comparative sea anemone stinging behavior. (**A**) *Nematostella vectensis* stings with tentacles while *Exaiptasia diaphana* also stings with acontia filaments that are ejected from its body for defense. *Left: Nematostella* nematocyte discharge was only observed in response to simultaneous prey chemicals and touch stimuli. *Middle, Right: Exaiptasia* nematocyte discharge from tentacles and acontia occurred irrespective of prey

*Figure 1 continued on next page*

*Figure 1 continued*

cues (touch alone). Scale bar = 50 µm. (**B**) *Nematostella* nematocyte discharge was elicited by simultaneous touch and prey chemical stimuli (n=10 trials). *Exaiptasia* tentacle (n=10) and acontia (n=13) nematocytes discharged only to touch, with or without prey chemicals. p<0.05 for *Nematostella*, paired two-tailed student's t-test. Data represented as mean ± sem.

To investigate whether different stinging behaviors might be suited for predation versus defense, we developed a normative theory aimed at predicting optimal stinging behavior as a function of nutritional state (see *Materials and methods* for model details). We focused on stinging intensity, defined as the fraction of nematocysts discharged during a stinging event, and asked whether nutritional state would affect optimal predatory and defensive stinging. In the language of decision models, the intensity of stinging is an action, and it is chosen by the agent, or anemone. In our study, 'choice' of stinging means modulation of behavior with nutritional state, rather than a cognitive process. Each choice has associated costs and benefits that depend on the environment. Because anemones sting many times over the course of their lives, an optimal behavior must account for overall costs and benefits after many events; therefore, this is a sequential decision-making problem.

We modeled the optimal stinging response to a given environment by using Markov decision processes (MDP). Each anemone was modeled as an 'agent' that must hedge the intensity of its stinging response. The environment, including the identity of prey, predators, and the physiological state of the animals, defines the likelihood, costs, and benefits of successful stinging. Specifically, intense stinging responses are costly since each fired nematocyst needs to be regenerated. But they are also more likely to succeed because greater discharge of stinging barbs increases the likelihood of contact and envenomation. The cost per nematocyte was first assumed to be constant and equivalent for defensive and predatory stinging as nematocyst discharge requires regeneration in either case (*Figure 2A*, solid line, filled circles). We assumed that the benefits of successful predatory stinging depend on the capture and consumption of prey, which improves satiation (*Figure 2B* **left**). In contrast, stinging a predator for defense would not improve nutritional state, hence the benefits of stinging would not depend on starvation (*Figure 2C* **left**). We then used the model to predict optimal stinging that maximizes the sum of all future benefits while minimizing costs during starvation (i.e. maximizing the value function, see Materials and methods). We then tested the prediction directly against experiments with behaving animals.

Using this approach, we found that optimal stinging strategies were completely different for predatory versus defensive behavior. Regardless of the specific environment (likelihood to succeed and specific costs and benefits), predatory stinging increased with starvation (*Figure 2B* **right,** solid lines, filled circles). To test our theory regarding predatory stinging, we carried out simulations in which agents discharged a random fraction of nematocytes between 0 and 1, regardless of starvation. Random stinging was unsustainable over numerous events and agents quickly reached maximal starvation state. Agents using optimal predatory stinging discharged more nematocysts when starved and less when satiated, leading to sustained stinging behavior and survival. This was true even if they fired the same fraction of nematocytes as the random agent (*Figure 2D*). In contrast, optimal stinging for defense stayed constant with starvation (*Figure 2C* **right,** solid lines, filled circles). Importantly, while the precise optimal response depended on the details of cost and reward that defined the MDP, the differences between increasing predatory stinging versus unchanging defensive stinging were consistent and largely independent of assumptions associated with each reward function (described in *Materials and methods*, *Figure 2—figure supplements 1–4*). These results reflect greater rewards to predatory anemones upon stinging during starvation, whereas defensive anemones sting at a similar rate regardless of nutritional status. Thus, our model predicts robust differences in predatory versus defensive stinging behavior.

We next sought to experimentally test whether pressure to predate regulates stinging in *Nematostella* and *Exaiptasia*. To do so, we fed both species of anemones copious amounts of prey (brine shrimp, *Artemia nauplii*) for 1–2 weeks and then deprived them of food for 5 days. Following manipulation of prey availability, *Nematostella* significantly increased stinging in response to starvation, while *Exaiptasia* stinging remained relatively constant despite complete deprivation of prey (*Figure 2E*, symbols with error bars). The behavior was remarkably consistent with our normative theory of optimal stinging strategies for predation versus defense (*Figure 2E*, filled circles). Furthermore, changes in

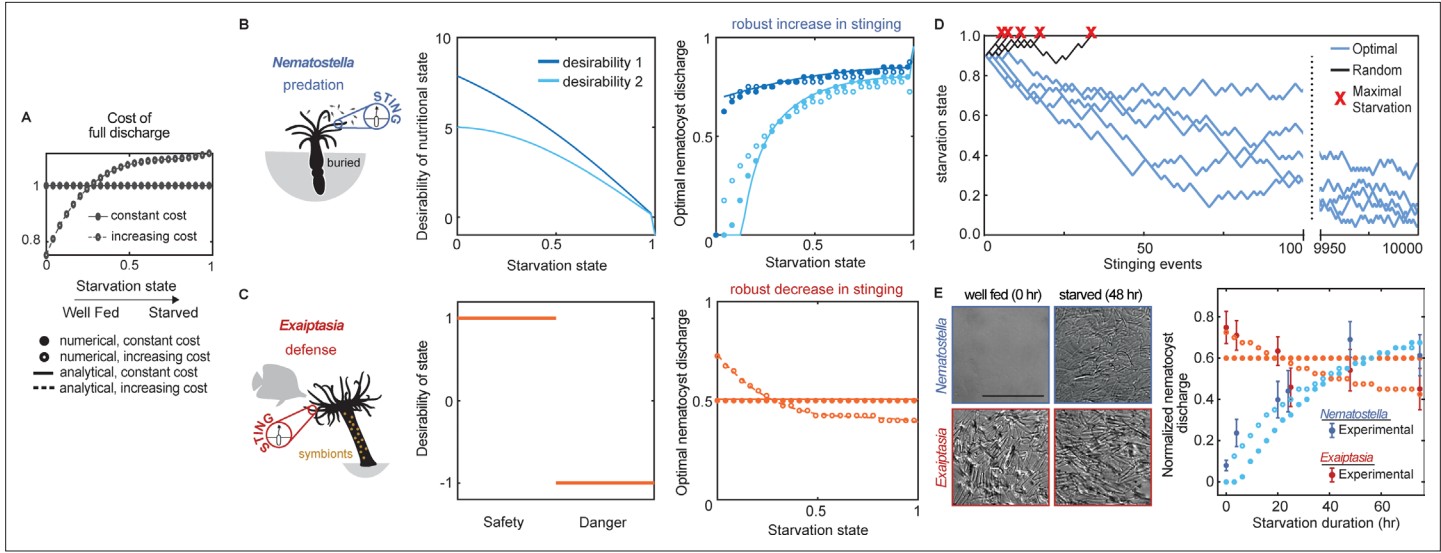

**Figure 2.** *Nematostella* stinging is regulated by predation while *Exaiptasia* stings for defense. (**A**) The cost of stinging is $c = c_o a$, where $c_o$ is the cost for full nematocyte discharge and it either does not change (solid lines filled circles) or increases slightly (dashed lines empty circles) with starvation state. These symbols are used throughout the figure to represent each cost function. The increasing cost is obtained by fitting the Exaiptasia behavior (*see fitting procedure in Materials and methods*). (**B**) *Left*: *Nematostella* burrows in the substrate and stings for predation. *Center*: Desirability of nutritional state, or reward, decreases with starvation. Two examples are shown: example 1, $r(s) = 10 \tan^{-1}(1 - s)$; example 2, $r(s) = 5 \cos\left(\frac{s\pi}{2}\right)$. *Right*: Predicted optimal stinging obtained by solving **equation (1)** with numerical simulations (circles) and approximate analytical solutions (lines) assuming: $p(a) = p_M a(2 - a)$ and $p_M = 0.8$; $c = c_0 a$ with cost for full discharge $c_0$ matching panel A (full circles and solid lines for constant cost; empty circles for increasing cost); reward in Left panels (colors match). For all reward and cost functions, optimal predatory stinging increases with starvation under broad assumptions (*see Materials and methods*). (**C**) *Left*: Exaiptasia diaphana relies heavily on endosymbiotic algae for nutrients and stings primarily for defense. *Center:* We assumed there are two states, safety (L), and danger (D). The state of safety can transition to danger, but not the other way around. We assumed the agent obtains reward 1 in state L and penalty –1 in state D. *Right*: Predicted optimal stinging obtained by solving **equation (2)** with numerical simulations (circles) and analytical solutions (lines). Styles match the costs in panel A; we assume $p(a) = p_M a(2 - a)$ and $p_M = 0.8$ as before. Optimal defensive stinging is constant or decreases with starvation under broad assumptions (*see Materials and methods*). (**D**) Examples of optimal (blue) versus random (black) predatory stinging. Each agent (anemone) starts with $s = 0.9$, and stings sequentially for many events (represented on the x axis). The random agent almost always reaches maximal starvation before 50 events (grey lines, five examples shown). In comparison, the optimal agent effectively never starves due to a successful stinging strategy optimized for predation (blue lines, five examples shown, parameters as in panel B, curve with matching color). (**E**) *Left*: *Nematostella* nematocyte discharge was affected by prey availability while *Exaiptasia* stung at a similar rate regardless of feeding. p<0.0001 for *Nematostella*, two-way ANOVA with post hoc Bonferroni test (n=10 animals, data represented as mean ± sem). *Right*: Experimental data (circles with error bars representing standard deviation) are well fit by normalized optimal nematocyst discharge predicted from MDP models for both *Exaiptasia* (orange full and empty circles for constant and increasing cost, panel A) and *Nematostella* (light blue full and empty circles for constant and increasing cost, panel A and desirability 2 in panel B). We match the last experimental data point to $s = 0.5$, the precise value of this parameter is irrelevant as long as it is smaller than 1, representing that animals are not severely starved during the experiment.

The online version of this article includes the following figure supplement(s) for figure 2:

**Figure supplement 1.** Sketch of Markov Decision Processes model and predictions for stinging.

**Figure supplement 2.** Optimal policy predicted by Bellman's theory for the MDP sketched in *Figure 2—figure supplement 1A*.

**Figure supplement 3.** Sketch of theoretical prediction for predatory stinging with increasing cost.

**Figure supplement 4.** Effects of a moderately *vs* dramatically increasing cost with starvation.

**Figure supplement 5.** Modulation of *Nematostella* and *Exaiptasia* stinging is not due to changes in the abundance of nematocytes.

Nematostella and Exaiptasia stinging were not due to changes in the abundance of nematocytes because tentacles from both animals were abundantly armed with nematocytes across feeding conditions (*Figure 2—figure supplement 5*). The experimental behavior of *Exaiptasia* showed a slight decrease in stinging with starvation. To account for this decrease we revisited the theory and assumed that the cost per nematocyte slightly increased with starvation (*Figure 2A*, dashed lines and open circles). In this case, the optimal response slightly decreased for defensive stinging but increased for predatory stinging (*Figure 2B* right, open circles and *Figure 2C* right, dashed lines, open circles). In fact, the fit between theory and data for both *Nematostella* and *Exaiptasia* improved when the cost increased slightly with starvation (*Figure 2E*, open circles). A more dramatic and less realistic increase

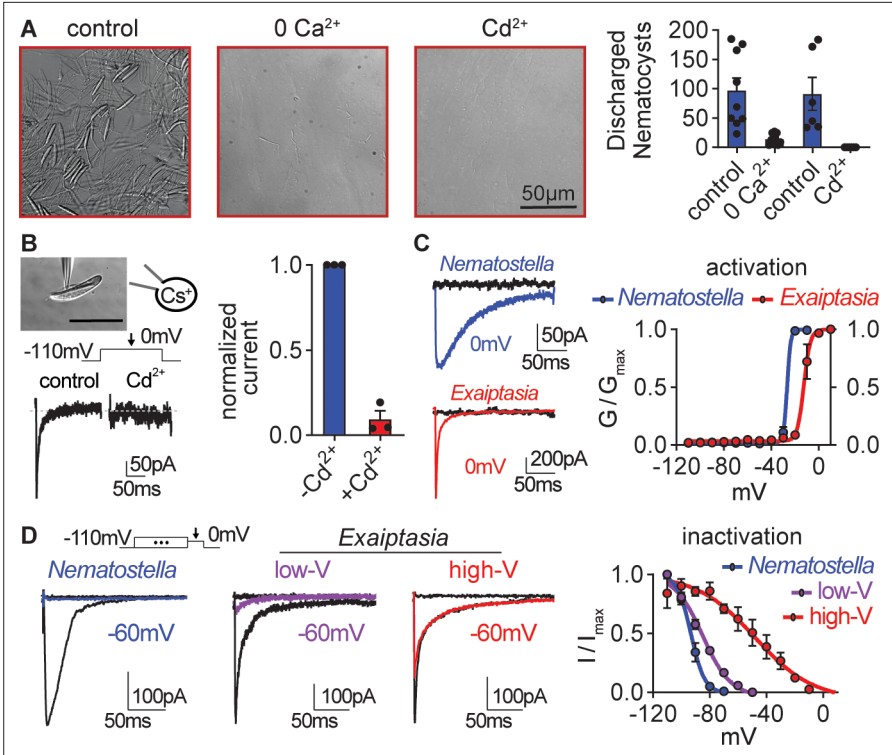

**Figure 3.** *Exaiptasia* nematocyte voltage-gated Ca$^{2+}$ currents exhibit minimal steady-state inactivation compared with *Nematostella*. (**A**) Touch-elicited *Exaiptasia* tentacle nematocyte discharge was blocked in the absence of Ca$^{2+}$ ($p<0.01$, paired two-tailed student's t-test, n=9 animals) or by addition of the Ca$_V$ channel blocker Cd$^{2+}$ (500 μM, $p<0.05$, paired two-tailed student's t-test, n=6 animals). Scale bar = 50 μm. (**B**) *Top:* Representative patch clamp experiment from an *Exaiptasia* nematocyte. Scale bar = 20 μm. *Bottom:* Nematocyte voltage-gated currents elicited by a maximally activating 0 mV pulse were blocked by Cd$^{2+}$ (n=3 cells, $p<0.01$, paired two-tailed student's t-test). (**C**) Nematocyte voltage-gated currents elicited by –120 mV (black) or 0 mV pulses (colored). Conductance-voltage curves for *Nematostella* nematocyte (V$_{a1/2}$ = -26.54 ± 0.78mV, n=3) and *Exaiptasia* nematocyte (V$_{a1/2}$ = -12.47 ± 0.70mV, n=3). (**D**) Nematocyte voltage-gated currents elicited by a maximally activating voltage pulse following 1 s pre-pulses to –110 mV (max current, black), –50 mV (colored), or 20 mV (inactivated, no current). *Nematostella* nematocytes inactivated at very negative voltages (V$_{i1/2}$ = -93.22 ± 0.42mV, n=7) while *Exaiptasia* contained two populations of nematocytes: low-voltage threshold (V$_{i1/2}$ = -84.94 ± 0.70mV, n=4), and high-voltage threshold (V$_{i1/2}$ = -48.17 ± 3.32mV, n=3). Data represented as mean ± sem.

of the cost with starvation may lead to a decrease in predatory stinging (*Figure 2—figure supplement 4*). Thus, we conclude that *Nematostella* controls stinging for opportunistic predation while *Exaiptasia* stinging is indiscriminate and serves a greater defensive role for this symbiotic anemone.

## Sea anemones with different stinging behavior use distinct Ca$_V$ channels

We next probed the physiological basis underlying these significantly different stinging behaviors. We previously found that *Nematostella* stinging is triggered by a specialized Ca$_V$ channel that exhibits strong inactivation at negative voltages to prevent responses to extraneous non-prey mechanical stimuli (*Weir et al., 2020*). Ca$^{2+}$ influx triggers an increase in hydrostatic pressure inside the nematocyst capsule that forces the stinging thread to evert explosively at an acceleration of up to $5.41×10^6$ g, placing it among the fastest biological processes in existence (*Lubbock and Amos, 1981*; *Lubbock et al., 1981*; *Holstein and Tardent, 1984*; *Weber, 1990*; *Gitter et al., 1994*; *Tardent, 1995*; *Nüchter et al., 2006*). Similar to *Nematostella*, *Exaiptasia* stinging required extracellular Ca$^{2+}$ and was abolished by Cd$^{2+}$, a Ca$_V$ channel blocker (*Figure 3A*). Consistent with a Ca$^{2+}$-dependent stinging mechanism, whole-cell patch clamp recordings from nematocytes revealed the presence of voltage-gated inward currents that were blocked by Cd$^{2+}$, suggesting that *Exaiptasia* nematocytes also use Ca$_V$ channels to

control stinging (**Figure 3B**). Indeed, $Ca_V$ currents in *Exaiptasia* nematocytes exhibited similar voltage-dependent activation properties compared with *Nematostella* nematocytes (**Figure 3C**). Thus, in agreement with previous findings, we conclude that $Ca^{2+}$ influx via $Ca_V$ channels is broadly important for stinging.

$Ca_V$ channels respond to positive membrane potentials by opening to conduct $Ca^{2+}$. However, sustained positive voltage drives $Ca_V$s to transition to a non-conducting state (inactivation) that prevents re-activation until channels return to a resting state induced by negative membrane potentials. In most cells, voltage-gated ion channel inactivation prevents extended responses to repetitive or prolonged stimulation. *Nematostella* $Ca_V$ is unusual because it inactivates at very negative voltages to prevent responses from resting potential, resulting in nematocytes that cannot fire from rest (**Weir et al., 2020**). In contrast to *Nematostella* nematocytes in which half of all $Ca_V$ channels ($V_{i1/2}$) were inactivated at ~ –93 mV, *Exaiptasia* nematocytes exhibited two distinct inactivation phenotypes: (1) nematocytes with low-voltage threshold (low-V) inactivation similar to that of *Nematostella* (low-V, $V_{i1/2}$ = ~ –85 mV); and (2) a distinct population with weak, high-voltage (high-V) threshold inactivation similar to its well-characterized mammalian orthologue (high-V, $V_{i1/2}$ = ~ –48 mV) (**Figure 3D**). While we did not observe a correlation with abundance or distinct cellular morphology (**Östman, 2000**; **Kass-Simon and Scappaticci, Jr., 2002**; **Grajales and Rodríguez, 2014**), we could clearly distinguish the two populations based on these electrophysiological features. Importantly, high-V nematocyte inactivation was minimal at resting voltages (~ –70 mV), so nearly all channels would be available to amplify depolarizing signals, such as those elicited by touch. Thus, these markedly different physiological properties correlate with distinct stinging behavior: *Nematostella* uses unusual low-voltage $Ca_V$ inactivation to integrate sensory cues for tightly regulated predatory stinging. In contrast, *Exaiptasia* employs a population of nematocytes with weak $Ca_V$ inactivation, consistent with direct activation from resting potentials and stinging to touch alone.

What is the molecular basis of distinct nematocyte physiology? $Ca_V$ channels are made of at least three subunits: the pore-forming α and auxiliary β and α2δ subunits. Transcriptomics revealed that nematocyte-enriched tentacles of *Exaiptasia* expressed *cacna1a*, the pore-forming subunit homologous to that of the previously characterized *Nematostella* nematocyte $Ca_V$ channel (**Figure 4—figure supplement 1**). We also analyzed the $Ca_V$ β subunit, $Ca_Vβ$, which is required and sufficient for the unusual inactivation properties observed in *Nematostella* $Ca_V$ (**Weir et al., 2020**; **Figure 4—figure supplement 1**). From *Exaiptasia*, we identified two isoforms of $Ca_Vβ$: $EdCa_Vβ1$ and $EdCa_Vβ2$. Droplet digital PCR assays of mRNA abundance showed that both isoforms are expressed throughout *Exaiptasia* tissues, suggesting they could both be functionally important (**Figure 4—figure supplement 1B**). To localize $Ca_Vβ$, we used in situ hybridization to determine that distinct nematocyte populations expressed either $EdCa_Vβ1$ or $EdCa_Vβ2$ mRNA (but not both) in the same cell (**Figure 4B**).

Considering this expression profile, we wondered if the two $Ca_Vβ$ isoforms could mediate low-V and high-V inactivation phenotypes in *Exaiptasia* nematocytes. To investigate this question, we heterologously expressed each β subunit isoform with other well-characterized $Ca_V$ subunits (mammalian $Ca_V$ α and α2δ) that express well in heterologous systems. Both channels exhibited functional $Ca_V$ currents with similar activation thresholds (**Figure 4C**). However, $EdCa_Vβ1$- and $EdCa_Vβ2$-containing channels significantly differed in their inactivation properties. $EdCa_Vβ1$ inactivated at negative voltages, similar to channels containing *Nematostella* $Ca_Vβ$ ($NveCa_Vβ$). In contrast, $EdCa_Vβ2$ mediated $Ca_V$ currents with weak inactivation, more like channels containing rat $Ca_Vβ2a$ (**Figure 4C and D**). Thus, $EdCa_Vβ1$ and $EdCa_Vβ2$ confer strong, low-voltage and weak, high-voltage steady-state inactivation, respectively, and are expressed in distinct nematocytes, consistent with low-V and high-V threshold inactivating nematocyte populations. Genomic alignment revealed that alternative splicing at the N-terminus gives rise to $EdCa_Vβ1$ and $EdCa_Vβ2$ isoforms, serving as a mechanism to dynamically tune nematocyte physiology and potentially stinging behavior in contrast to adaptation through gene duplication and divergence (**Figure 4E**). Furthermore, by expressing two functional variants, *Exaiptasia* could use distinct nematocyte populations for different behaviors, including a less pronounced role for predation.

## Structural adaptations across cnidarian $Ca_V$ channels

We next asked how variation in $Ca_Vβ$ structure mediates strong phenotypes by testing whether distinct protein domains confer low or high voltage-dependent inactivation. We first compared rat

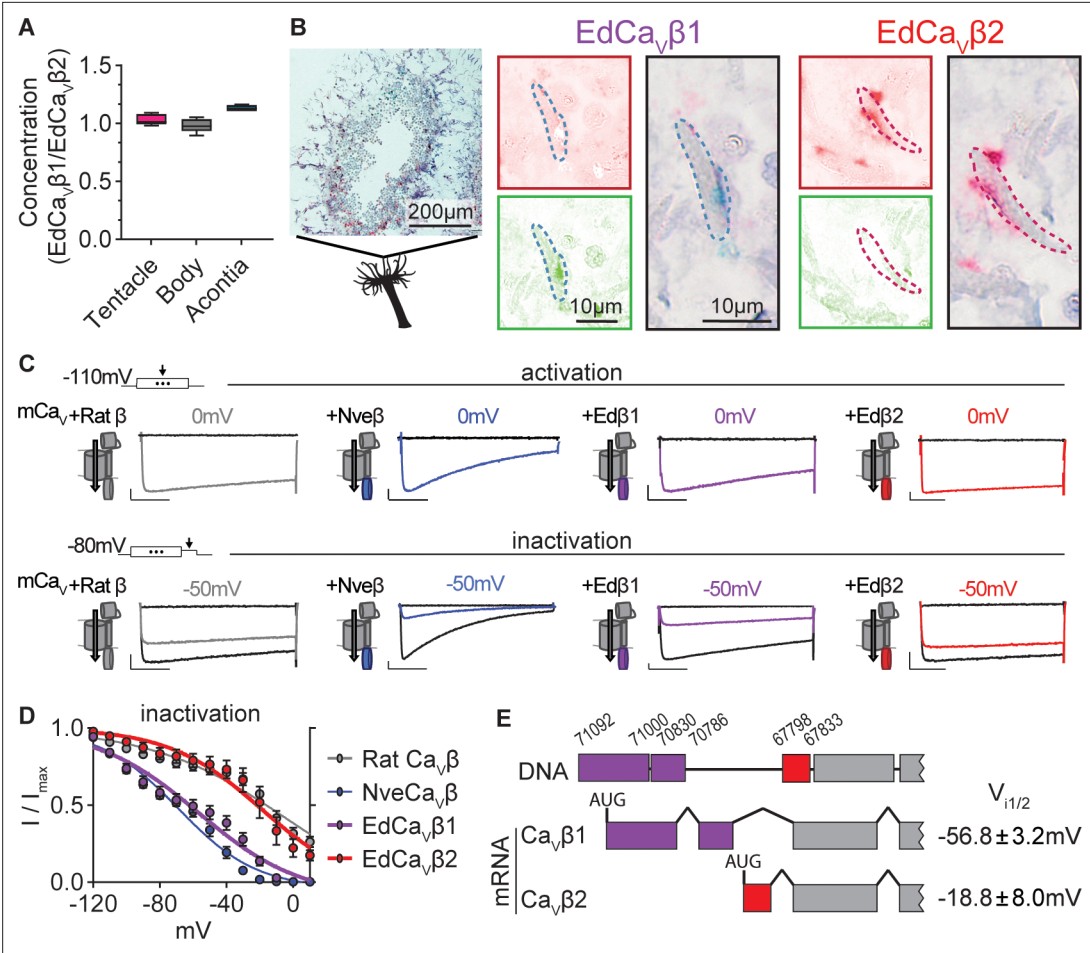

**Figure 4.** *Exaiptasia* expresses a Ca$_V$ β subunit splice isoform that confers weak voltage-dependent inactivation. (**A**) ddPCR ratio of concentrations of Ca$_V$ β subunit 1 and 2 mRNAs was similar in tentacle (n=5), body (n=5), and acontia (n=4 animals) tissue samples. (**B**) EdCa$_V$β1 and EdCa$_V$β2 localized to distinct nematocytes in *Exaiptasia* tentacle cross section, as visualized by BaseScope in situ hybridization. Representative nematocyte expressing EdCa$_V$β1 (green) or EdCa$_V$β2 (red). Representative of three animals. (**C**) Voltage-gated currents from heterologously-expressed chimeric mammalian Ca$_V$ (mCa$_V$) with different β subunits: rat (*Rattus norvegicus*), *Nematostella* (Nve), *Exaiptasia* EdCa$_V$β1 or EdCa$_V$β2. *Top*: Currents elicited by voltage pulses to –120 mV (no current, black) and maximally activating 0 mV (colored). *Bottom*: Voltage-gated currents elicited by a maximally activating voltage pulse following 1 s pre-pulses to –110 mV (max current, black), –50 mV (colored), or 20 mV (inactivated, no current, black). Scale bars = 100 pA, 50ms. (**D**) *Exaiptasia* Ca$_V$ β subunit splice isoforms confer distinct inactivation: *Nematostella* β subunit (V$_{1/2}$ = -68.93 ± 1.53mV, n=5) and Rat β2a subunit (V$_{1/2}$ = -2.98 ± 13.51mV, n=12) and EdCa$_V$β1 (V$_{1/2}$ = -56.76 ± 3.18mV, n=8), and EdCa$_V$β2 (V$_{1/2}$ = -18.84 ± 8.00mV, n=5 cells). Data represented as mean ± sem. (**E**) Genomic alignment of *Exaiptasia* β subunit isoforms showed that alternative splicing of the N-terminus region was associated with distinct inactivation: Ca$_V$β1 (long N-term) had low-voltage steady-state inactivation similar to *Nematostella*, while Ca$_V$β2 (short N-term) exhibited more depolarized steady-state inactivation, matching its mammalian orthologue. Genomic loci listed above sequence.

The online version of this article includes the following figure supplement(s) for figure 4:

**Figure supplement 1.** Transcriptomic and molecular analyses of *Exaiptasia* β subunit isoforms.

rCavβ2a and *Nematostella* NveCa$_V$β, which have significantly different voltage-dependent properties (***Weir et al., 2020***). Swapping the well-characterized SH3, HOOK, and GK domains had no effect on inactivation, but the NveCa$_V$β N-terminus was both required and sufficient for low voltage-dependent inactivation (***Figure 5A and B***). Indeed, swapping only the N-terminus of NveCa$_V$β was sufficient to shift rat rCavβ2a-conferred inactivation by ~ –75 mV (***Figure 5A and B***). This finding is consistent with the variation in EdCa$_V$β splice isoforms, in which differences in the N-terminus account for a~40 mV difference in inactivation thresholds.

To explore evolutionary relationships of Ca$_V$β, we constructed a phylogenetic tree of sequences from various cnidarians including *Nematostella vectensis* (anemone, NveCa$_V$β), *Exaiptasia diaphana* (anemone, EdCa$_V$β1 and EdCa$_V$β2), *Cyanea capillata* (jellyfish, CcCa$_V$β), *Physalia physalis* (hydrozoan,

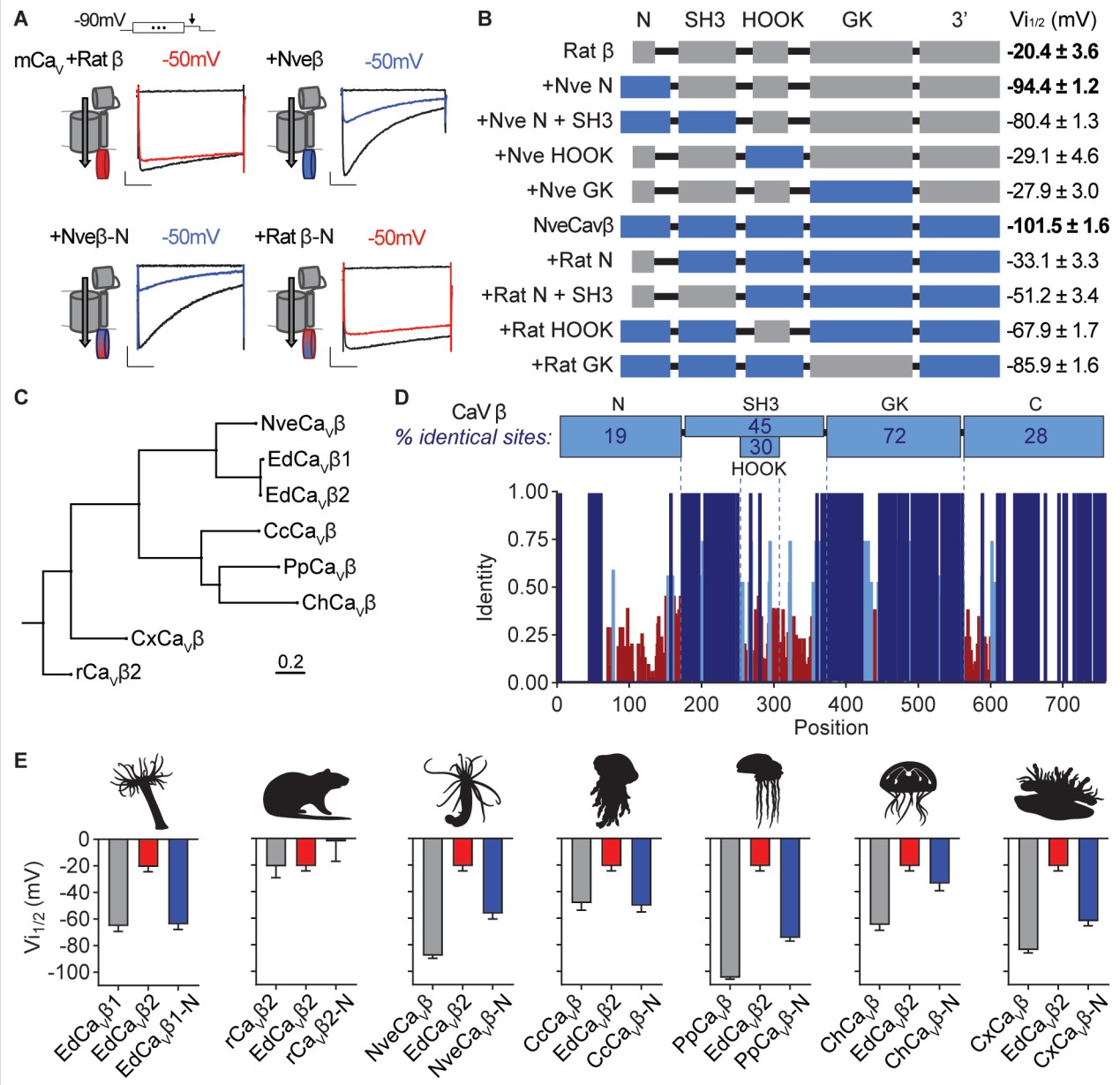

**Figure 5.** Cnidarian $Ca_V$ β subunit N-termini confer unique inactivation properties. (**A**) Voltage-gated currents from heterologously expressed $Ca_V$ channels with *Nematostella*-rat chimeric β subunits demonstrate that the *Nematostella* N-terminus is sufficient to drive inactivation at negative voltages. Currents shown in response to 10 mV voltage pulses following 1 s pre-pulses to −130 mV (max current, black), −50 mV (colored), or 0 mV (inactivated, no current, black). Scale bars = 100 pA, 50ms. (**B**) Diagram of $Ca_V$ *Nematostella*-rat β subunit domain swaps and resulting $V_{i1/2}$ values. The *Nematostella* β subunit N-terminus is required and sufficient for uniquely hyperpolarized $Ca_V$ inactivation properties (p<0.001 for average $V_{i1/2}$ values across mutant beta subunits, one-way ANOVA with post-hoc Tukey test, n=2–8 cells). (**C**) Phylogenetic tree of β subunit sequences obtained from several species of cnidarians. Abbreviations of species: Nve, *Nematostella vectensis*; Ed, *Exaiptasia diaphana*; Cc, *Cyanea capillata* (jellyfish); Pp, *Physalia physalis* (siphonophore); Ch, *Clytia hemisphaerica* (jellyfish); Cx, *Cassiopea xamachana* (jellyfish); r, *Rattus norvegicus*. (**D**) *Top*: Percentage of identity between amino acid sequences across β subunit protein domains for NveCa_Vβ, EdCa_Vβ1, EdCa_Vβ2, CcCa_Vβ, PpCa_Vβ, ChCa_Vβ, CxCa_Vβ2, rCa_Vβ2. *Bottom*: Fraction of identity of amino acids across sites of the β subunit protein. (**E**) Cnidarian $Ca_V$ β N-termini shift depolarized, weak voltage-dependent inactivation of $Ca_V$ channels containing EdCa_Vβ2 to more negative voltages. Voltage-dependent inactivation ($V_{i1/2}$) of heterologously-expressed $Ca_V$s with WT EdCa_Vβ2, β subunits from the indicated cnidarians, and chimeras with their N-termini on EdCa_Vβ2 (p<0.0001 for average $V_{i1/2}$ values with multiple comparisons against WT EdCa_Vβ2 mean, one-way ANOVA with Bartlett's test and post-hoc Tukey test, n=4–9 cells). Data represented as mean ± sem.

The online version of this article includes the following source data and figure supplement(s) for figure 5:

**Source data 1.** Amino acid sequences of beta subunits.

**Figure supplement 1.** Voltage-dependent activation of $Ca_V$ channels is conserved across cnidarian β subunits.

PpCa$_V$β), *Clytia hemisphaerica* (jellyfish, ChCa$_V$β), *Cassiopea xamachana* (jellyfish, CxCa$_V$β), and the Rat β subunit (rCavβ2a) as an outgroup (*Figure 5C*). Sequence comparison across all amino acid positions revealed that the N-terminus exhibited the greatest sequence diversity (*Figure 5D*), consistent with previous findings showing extensive alternative splicing in this region in other organisms (*Helton and Horne, 2002*; *Helton et al., 2002*; *Takahashi et al., 2003*; *Foell et al., 2004*; *Vendel et al., 2006*; *Ebert et al., 2008*; *Buraei and Yang, 2010*; *Siller et al., 2022*). We found that all cnidarian Ca$_V$βs conferred voltage-gated currents when co-expressed with Ca$_V$α and α2δ subunits and had relatively low voltage thresholds for inactivation compared with rCavβ2a or EdCa$_V$β2 (*Figure 5E*, *Figure 5— figure supplement 1A, B*). Importantly, swapping the N-termini of each cnidarian Ca$_V$β onto EdCa$_V$β2 was sufficient to shift voltage-dependent inactivation to more negative values (*Figure 5E*, *Figure 5— figure supplement 1C*). Thus, alternative splicing at the N-terminus could serve as a broad molecular mechanism for tuning Ca$_V$ function. Collectively, these findings substantiate the importance of Ca$_V$β in modulating inactivation and suggest a mechanism that could dynamically regulate a small region of only one subunit in the Ca$_V$ protein complex to tune complex stinging behavior.

## Discussion

Collectively, our studies on cnidarian stinging, here and (*Weir et al., 2020*), reveal different behavior in the primarily predatory anemone *Nematostella* versus the symbiotic anemone *Exaiptasia*. This study used a combination of theory and experimentation to uncover the molecular basis of regulation of the divergent behavior of *Exaiptasia* that uses stinging primarily for defense. Indeed, *Exaiptasia* obtains a large fraction of its energy and nutrients from endosymbiotic algae (*Muscatine et al., 1981*; *Shick and Dykens, 1984*; *Steen, 1988*), thus reducing overall pressure to predate. This finding is consistent with a common ecological theme in which symbiotic relationships are established whereby one partner provides food and the other provides shelter and defense (*Lehnert et al., 2012*; *Bucher et al., 2016*). Therefore, it is plausible that synergistic selection drives higher investment in defensive structures to protect symbiotic species.

Our results demonstrate that molecular adaptations tune distinct stinging behavior: *Nematostella* Ca$_V$β confers an unusually low threshold for inactivation, basally inhibiting nematocytes unless they are exposed to synergistic prey cues: chemical (hyperpolarizing to relieve inactivation) and mechanical (depolarizing to recruit available Ca$_V$ channels and elicit stinging; *Weir et al., 2020*). These physiological mechanisms reflect a stinging strategy suited to opportunistic predation by *Nematostella*, which burrow within shallow marshes and sting unsuspecting prey. Consistent with the predictions of optimal control theory, *Nematostella* increased stinging with starvation, suggesting that evolution has shaped its stinging response to maximize benefits for predation. In contrast, *Exaiptasia* nematocytes contain a functionally specialized splice variant of Ca$_V$β to mediate high threshold voltage-dependent Ca$_V$ inactivation, consistent with Ca$_V$ channel availability to amplify depolarizing signals from rest and stinging in response to touch alone. Thus, *Exaiptasia* physiology is consistent with an indiscriminate stinging strategy for defense, necessary for survival in an exposed environment that facilitates endosymbiotic photosynthesis (*Muscatine et al., 1981*; *Shick and Dykens, 1984*; *Steen, 1988*). Such stinging behavior is likely synergistic with physical escape for some cnidarians (*Pallasdies et al., 2019*; *Wang et al., 2023*). Consistent with the predictions of optimal control theory, *Exaiptasia* stinging was nearly independent of starvation, suggesting that evolution has shaped the stinging response to maximize benefits for defense. Using molecular information gleaned from analyzing these two cnidarians, we find that Ca$_V$β variation across cnidarians mediates differences in voltage-dependent inactivation, which could contribute to differences in stinging behavior. Thus, our study provides an example by which alternative splicing could account for adaptation across this diverse plethora of organisms and habitats.

While theory predicts robust trends for optimal predation and defense independent of environment, the precise nature of the predicted behavior does depend on the environment. In vivo, stinging is likely influenced by stimulus identity and intensity, background turbulence, and other factors. For example, cnidarians may have evolved distinct innate responses for different prey and use chemical sensing to enact the appropriate stinging response. In this case, optimal control theory can be used to predict the optimal response to known salient environmental cues. Alternatively, cnidarians may learn that specific prey are palatable and easy to catch through repeated exposure (*Botton-Amiot et al., 2023*). In this example, optimal control theory must be replaced by reinforcement learning as the

likelihood of successful predation and its cost and benefits (the environment) are unknown (*Sutton, 2018*).

Indeed, stinging is a complex process mediated by numerous molecular components and cell types that could be subject to evolutionary change or acute modulation. Cnidarians occupy diverse ecological niches and experience varying metabolic demands, predatory challenges, and other survival pressures that could influence stinging behavior. Beyond cnidarians with stationary lifestyles that support photosynthetic endosymbionts (symbiotic anemones, corals, sea pens) and those that use opportunistic 'sit-and-wait' ambush predatory strategies (burrowing anemones, siphonophores), others have evolved mobile lifestyles to actively capture prey (jellyfish) (*Muscatine et al., 1981*; *Shick and Dykens, 1984*; *Steen, 1988*; *Fraune et al., 2016*; *Damian-Serrano et al., 2022*). Stinging by mobile cnidarians could be subject to different physical demands, such as mechanical disturbance from increased turbulence that could necessitate distinct molecular control. Furthermore, stinging can be influenced by acute factors such as physiological state and various sensory cues, including chemicals, touch, or light (*Pantin, 1942*; *Giebel et al., 1988*; *Thorington and Hessinger, 1988*; *Watson and Hessinger, 1989*; *Plachetzki et al., 2012*; *Ozment et al., 2021*; *Aguilar-Camacho et al., 2023*). Thus, further inquiry into modulation of stinging across physiological states such as nutritional condition, altered symbiotic relationships, or developmental stages (*Sandberg et al., 1971*; *Columbus-Shenkar et al., 2018*) could reveal dynamic regulation by synaptic connections, hormones, or modulation of transcriptional or translational programs (*Westfall et al., 1998*; *Westfall et al., 2002*; *Westfall, 2004*; *Weir et al., 2020*). Importantly, across all these scenarios, nematocytes remain single-use cells, so it is essential that signaling cascades control discharge in response to the most salient environmental stimuli.

As an early-branching metazoan lineage and sister group to Bilateria (*Cartwright et al., 2007*), cnidarians are a useful model for probing origins of the nervous system and behavioral specialization (*Steele et al., 2011*; *Jékely et al., 2015*; *Pallasdies et al., 2019*). Here, we present a comparative approach across related cnidarians with distinct physiology and ecology to suggest that behavioral complexity emerges from subtle tuning of single proteins, even in non-neuronal cells. Indeed, cnidarians pose a unique opportunity for the integrative exploration of the evolution of animal behavior. Even beyond neural computations, the emergence of novel cell types among diverse cnidarian body plans, sophisticated predator-prey interactions, and symbioses all contribute to biological novelty and niche expansion (*Technau and Steele, 2012*). Overall, this work demonstrates how studying evolutionary novelties like stinging behavior can yield broad insight into signal transduction, cellular decision making, and suggests that the evolution of behavior should be examined across all tiers of biological organization.

# Materials and methods

## Key resources table

| Reagent type (species) or resource | Designation | Source or reference | Identifiers | Additional information |
|---|---|---|---|---|
| Strain, strain background (*Nematostella vectensis*, adult, male and female) | Nematostella | Woods Hole Marine Biological Laboratory | | |
| Strain, strain background (Exaiptasia diaphana, adult, male and female) | Exaiptasia | Carolina Biological Supply Company | Item #162865 | https://www.carolina.com/marine-and-saltwater-animals/sea-anemone-aiptasia-living/162865.pr |
| Strain, strain background (Cassiopea spp., adult, male and female) | Cassiopea xamachana | Carolina Biological Supply Company | Item #162936 | https://www.carolina.com/marine-and-saltwater-animals/mangrove-jellyfish-living-pack-of-5/162936.pr?question=mangrove |
| Cell line (Homo-sapiens) | HEK293T | ATCC | | CRL-3216 |
| Recombinant DNA reagent | nCa$_V$: cacnb2 (plasmid) | *Weir et al., 2020* | | NveCa$_V$β |
| Recombinant DNA reagent | Rat cacna2d1 | *Lin et al., 2004* | Addgene Plasmid #26575 | rCa$_V$α2δ |
| Recombinant DNA reagent | Mouse cacna1a | *Richards et al., 2007* | Addgene Plasmid #26578 | mCa$_V$α |
| Recombinant DNA reagent | Rat cacnb2a | *Wyatt et al., 1998* | Addgene Plasmid #107424 | rCa$_V$β |
| Recombinant DNA reagent | Exaiptasia diaphana cacnb1 (plasmid) | This paper | | EdCa$_V$β1 |
| Recombinant DNA reagent | Exaiptasia diaphana cacnb2 (plasmid) | This paper | | EdCa$_V$β2 |

*Continued on next page*

*Continued*

| Reagent type (species) or resource | Designation | Source or reference | Identifiers | Additional information |
|---|---|---|---|---|
| Recombinant DNA reagent | Physalia physalis cacnb (plasmid) | This paper | | PpCa$_v$β |
| Recombinant DNA reagent | Cyanea capillata cacnb (plasmid) | This paper | | CcCa$_v$β |
| Recombinant DNA reagent | Cassiopea xamachana cacnb (plasmid) | This paper | | CxCa$_v$β |
| Recombinant DNA reagent | Clytia hemisphaerica cacnb (plasmid) | This paper | | ChCa$_v$β |
| Recombinant DNA reagent | EdCa$_v$β2 with NveCa$_v$β NTerm (plasmid) | This paper | | NveCa$_v$β-N |
| Recombinant DNA reagent | EdCa$_v$β2 with rCa$_v$β NTerm (plasmid) | This paper | | rCa$_v$β-N |
| Recombinant DNA reagent | EdCa$_v$β2 with EdCa$_v$β1 NTerm (plasmid) | This paper | | EdCa$_v$β1-N |
| Recombinant DNA reagent | EdCa$_v$β2 with PpCa$_v$β NTerm (plasmid) | This paper | | PpCa$_v$β-N |
| Recombinant DNA reagent | EdCa$_v$β2 with CcCa$_v$β NTerm (plasmid) | This paper | | CcCa$_v$β-N |
| Recombinant DNA reagent | EdCa$_v$β2 with CxCa$_v$β NTerm (plasmid) | This paper | | CxCa$_v$β-N |
| Recombinant DNA reagent | EdCa$_v$β2 with ChCa$_v$β NTerm (plasmid) | This paper | | ChCa$_v$β-N |
| Recombinant DNA reagent | NveCa$_v$β with Mouse Ca$_v$β HOOK domain (plasmid) | This paper | | NVE_cacnb2_mus_hook |
| Recombinant DNA reagent | Mouse Ca$_v$β with NveCa$_v$β HOOK domain (plasmid) | This paper | | Mus_cacnb2_NVE_hook |
| Recombinant DNA reagent | NveCa$_v$β with Mouse Ca$_v$β GK domain (plasmid) | This paper | | NVE_cacnb2_mus_GK_domain |
| Recombinant DNA reagent | Mouse Ca$_v$β with NveCa$_v$β GK domain (plasmid) | This paper | | Mus_cacnb2_NVE_GK_domain |
| Recombinant DNA reagent | NveCa$_v$β with rCa$_v$β NTerm domain (plasmid) | This paper | | Rat_Nterm_NVE_cacnb2a |
| Recombinant DNA reagent | NveCa$_v$β with rCa$_v$β NTerm and SH3 domains (plasmid) | This paper | | Rat_Nterm_SH3_NVE_cacnb2a |
| Recombinant DNA reagent | rCa$_v$β with NveCa$_v$β NTerm and SH3 domains domain (plasmid) | This paper | | NVE_Nterm_SH3_rat_cacnb2a |
| Recombinant DNA reagent | rCa$_v$β with NveCa$_v$β NTerm domain (plasmid) | This paper | | NVE_Nterm_rat_cacnb2a |
| Sequence-based reagent | EdCa$_v$β1 primer F | This paper | Custom PCR primer | GGATTTCGCCCTGAGCAA |
| Sequence-based reagent | EdCa$_v$β1 primer R | This paper | Custom PCR primer | TGGATCTCCGAAGGAGTTGA |
| Sequence-based reagent | EdCa$_v$β1 probe | This paper | Custom probe | FAM-ACATAGAACTTGATAGTCTCGAGCACG-BHQ-1 |
| Sequence-based reagent | EdCa$_v$β2 primer F | This paper | Custom PCR primer | TGTCCAGAGCTTCACAAAGG |
| Sequence-based reagent | EdCa$_v$β2 primer R | This paper | Custom PCR primer | TCTTCGTCAACACTACTTGTATCA |
| Sequence-based reagent | EdCa$_v$β2 probe | This paper | Custom probe | HEX-ACTCAGAACCTGCTTACAGAGCCT-BHQ-1 |
| Commercial assay or kit | One-Step RT-ddPCR Advanced Kit for Probes | Bio-Rad Laboratories | Cat #1864021 | |
| Commercial assay or kit | Droplet Gen Oil for Probes | Bio-Rad Laboratories | Cat #1863005 | |
| Chemical compound, drug | CdCl$_2$ | Sigma | Cat #202908 | |
| Software, algorithm | Rcorrector | *Song and Florea, 2015* | | https://github.com/mourisl/Rcorrector |
| Software, algorithm | FilterUncorrectablePEfastq.py | Harvard Informatics | | https://github.com/harvardinformatics/TranscriptomeAssemblyTools |
| Software, algorithm | Trimgalore | The Babraham Institute Bioinformatics Group | | https://github.com/FelixKrueger/TrimGalore |
| Software, algorithm | Bowtie2 | *Langmead and Salzberg, 2012* | | https://bowtie-bio.sourceforge.net/bowtie2/index.shtml |

*Continued on next page*

*Continued*

| Reagent type (species) or resource | Designation | Source or reference | Identifiers | Additional information |
|---|---|---|---|---|
| Software, algorithm | Fastqc | The Babraham Institute Bioinformatics Group | | https://www.bioinformatics.babraham.ac.uk/projects/fastqc/ |
| Software, algorithm | Trinity | *Grabherr et al., 2011* | | https://github.com/trinityrnaseq |
| Software, algorithm | Transdecoder | Haas, BJ. https://github.com/ TransDecoder/TransDecoder | | https://github.com/TransDecoder/TransDecoder |
| Software, algorithm | DIAMOND | *Buchfink et al., 2015* | | https://github.com/bbuchfink/diamond |
| Software, algorithm | Kallisto | *Bray et al., 2016* | | https://github.com/pachterlab/kallisto |
| Software, algorithm | Clustal Omega | *Madeira et al., 2022* | | https://www.ebi.ac.uk/Tools/msa/clustalo/ |
| Software, algorithm | Geneious Prime | Geneious | RRID: SCR_010519 | https://www.geneious.com |
| Software, algorithm | MAFFT v.7 | *Katoh and Standley, 2013* | | https://mafft.cbrc.jp/alignment/server/ |
| Software, algorithm | ModelFinder | *Kalyaanamoorthy et al., 2017* | | http://www.iqtree.org/ModelFinder/ |
| Software, algorithm | IQ-TREE v2.0 | *Minh et al., 2020* | | http://www.iqtree.org/ |
| Software, algorithm | UFBoot2 | *Hoang et al., 2018* | | https://bio.tools/ufboot2 |
| Software, algorithm | QuantaSoft | Bio-Rad Laboratories | | https://www.bio-rad.com/en-us/life-science/digital-pcr/qx200-droplet-digital-pcr-system/quantasoft-software-regulatory-edition |
| Software, algorithm | pClamp 11 Software Suite | Molecular Devices | | https://www.moleculardevices.com/products/axon-patch-clamp-system/acquisition-and-analysis-software/pclamp-software-suite |
| Software, algorithm | MetaMorph Microscopy Automation and Image Analysis Software | Molecular Devices | | https://www.moleculardevices.com/products/cellular-imaging-systems/acquisition-and-analysis-software/metamorph-microscopy |
| Software, algorithm | Fiji (ImageJ) | *Schindelin et al., 2012* | RRID: SCR_002285 | |
| Software, algorithm | GraphPad Prism | GraphPad Software | RRID: SCR_002798 | http://www.graphpad.com/ |
| Software, algorithm | Cell Sens | Olympus | | https://www.olympus-lifescience.com/en/software/cellsens/ |
| Commercial assay or kit | BaseScope Duplex Reagent Kit | Advanced Cell Diagnostics | Cat #323800 | |
| Sequence-based reagent | BaseScope Probe-BA-Epa-LOC110232233-tvX3-1zz-st-C1 (EdCa$_v$β1) | Advanced Cell Diagnostics | Custom probe 1162801 C1 | |
| Sequence-based reagent | BaseScope Probe-BA-Epa-LOC110232233-tvX6-1zz-st-C2 (EdCa$_v$β2) | Advanced Cell Diagnostics | Custom probe 1162811 C2 | |
| Sequence-based reagent | BaseScope Duplex Positive Control Probe-Human [Hs]-C1-Hs-PPIB-1zz/C2-POLR2A-1zz | Advanced Cell Diagnostics | Cat #700101 | |
| Sequence-based reagent | BaseScope Duplex Negative Control Probe-C1-DapB-1zz/C2-DapB-1zz | Advanced Cell Diagnostics | Cat #700141 | |

## Animals and cells

Starlet sea anemones (*Nematostella vectensis*) were provided by the Marine Biological Laboratory (Woods Hole, Massachusetts). Adult animals of both sexes were used and kept on a 14 hr light/10 hr dark cycle at 26 °C in 1/3 natural sea water (NSW). *Exaiptasia spp.* were purchased through Carolina Biological Supply Company (Cat #162865). Adult animals of both sexes were used following being kept on either a 10 hr light/14 hr dark cycle at 26 °C in natural sea water (NSW) or a 14 hr light/10 hr dark cycle at 26 °C in natural sea water (NSW). *Cassiopea spp.* were purchased through Carolina Biological Supply Company (Cat #162936). Unless stated otherwise, all animals were fed freshly hatched brine shrimp (Artemia) twice a week.

*Exaiptasia diaphana* were bleached through chemical methods (menthol-induced). Menthol (100 mM in 100% ethanol; Sigma-Aldrich) was added to NSW at a final concentration of 0.2 mM (*Matthews et al., 2016*). The anemones were incubated in the menthol/NSW treatment solution for a maximum of 8 hr per day and outside of treatments anemones were incubated in NSW. For 2 weeks, anemones were treated 4 days per week and kept in the dark continuously starting from day 1 of treatment, aside from treatment changes. Animals were fed with Artemia approximately twice per week

between bleaching treatments, enabling successful bleaching with minimal mortality. Their symbiotic status was assessed via fluorescence microscopy at the end of each week. For starvation experiments, animals were fed to excess for 1–2 weeks before the trial and withheld food entirely during the trial period and given water changes twice a week.

*Nematostella* nematocytes were isolated from tentacle tissue, which was harvested by anesthetizing animals in high magnesium solution containing (mM): 140 NaCl, 3.3 Glucose, 3.3 KCl, 3.3 HEPES, 40 MgCl$_2$. Cells were isolated from tentacles immediately prior to electrophysiology experiments by treatment with 0.05% Trypsin at 37 °C for 15–20 min and mechanical dissociation in divalent free recording solution (mM): 140 NaCl, 3.3 Glucose, 3.3 KCl, 3.3 HEPES, pH 7.6. Dissociated cells were held on ice until use. Basitrichous isorhiza nematocytes were isolated from tentacles and identified by the presence of a capsule with high refractive index containing a barbed thread, oblong shape, and the presence of a cnidocil. *Exaiptasia* nematocytes were isolated from tentacle tissue immediately prior to electrophysiology experiments by incubation in a heat shock dissociation solution with (in mM): 430 NaCl, 10 KCl, 150 sucrose, 5 NaEGTA, 10 HEPES, 10 glucose, pH 7.6 at 45 °C for 15–20 min and mechanical dissociation in the same solution. Dissociated cells were held on ice until use. Nematocytes were isolated from tentacles and identified by the presence of a capsule with high refractive index, oblong shape, and the presence of one or multiple apical cilia.

HEK293T cells (ATCC, Cat# CRL-3216, RRID:CVCL_0063, authenticated and validated as negative for mycoplasma by vendor) were grown in DMEM, 10% fetal calf serum, and 1% penicillin/streptomycin at 37 °C, 5% CO$_2$. For transfection, HEK293 cells were washed with Opti-MEM Reduced Serum Media (Gibco) and transfected using lipofectamine 2000 (Invitrogen/Life Technologies Cat #11668019) according to the manufacturer's protocol. One µg each of *M. musculus* (mouse) *cacna1a* and rat *cacna2d1* and one of a wide variety of beta subunits (*Nematostella vectensis cacnb2.1* (NveCa$_V$β), *Rattus norvegicus* (rat) *cacnb2a* (rCa$_V$β2a), *Exaiptasia diaphana* Ca$_V$βs (EdCa$_V$β1, EdCa$_V$β2), *Cyanea capillata* Ca$_V$β (CcCa$_V$β), *Physalia physali*s Ca$_V$β (PpCa$_V$β), *Clytia hemisphaerica* Ca$_V$β (ChCa$_V$β), *Cassiopea xamachana* Ca$_V$β (CxCa$_V$β2)) were coexpressed with 0.5 µg eGFP. We also assayed an array of different EdCa$_V$β2 mutants with N-termini from different animals by coexpressing 0.5 µg eGFP, 1 µg of *M. musculus* (mouse) *cacna1a* and rat *cacna2d1,* and one of a variety of beta subunits (*Nematostella vectensis cacnb2.1* mutant (NveCa$_V$β-N), *R. norvegicus* (rat) *cacnb2a* mutant (rCa$_V$β-N), *Exaiptasia diaphana* Ca$_V$β mutants (EdCa$_V$β1-N, EdCa$_V$β2-N), *Cyanea capillata* Ca$_V$β mutant (CcCa$_V$β-N), *Physalia physali*s Ca$_V$β mutant (PpCa$_V$β-N), *Clytia hemisphaerica* Ca$_V$β mutant (ChCa$_V$β-N), *Cassiopea xamachana* Ca$_V$β mutant (CxCa$_V$β2-N)). To enhance channel expression, cells were incubated with transfection mix containing plasmid DNA and Lipofectamine 2000 in Opti-MEM for 6 hr at 37 °C. Cell were then re-plated on coverslips, incubated for 1–2 hr at 37 °C, and then incubated at 30 °C for 2–6 days before experiments. Rat *cacna2d1* (RRID: Addgene_26575) and *cacna1a* were gifts from D. Lipscombe (RRID: Addgene_26578) and *cacnb2a* was a gift from A. Dolphin (RRID: Addgene_107424).

## Molecular biology

RNA was prepared from tentacles, body, and acontia tissues of WT and bleached adult *Exaiptasia* using published methods (*Stefanik et al., 2013*). Each tissue was homogenized (Millipore Sigma Cat #Z359971) and RNA was extracted using TRIzol Reagent (Thermo Fisher Cat #15596026), then after skipping the salt precipitation steps, RNA was purified and concentrated with the RNA Clean & Concentrator-5 kit (Zymo Research). For ddPCR experiments, droplet generation (QX200 Droplet Generator BioRad Cat #1864002) and transfer of droplets to ddPCR 96-Well Plates (Bio-Rad Cat #12001925) were performed according to manufacturer's instructions (Instruction Manual, QX200 Droplet Generator – Bio-Rad). Custom primers and probes and One-Step RT-ddPCR Advanced Kit for Probes (Bio-Rad Cat #1864021) reaction reagents and Droplet Generation Oil for Probes (Bio-Rad Cat #1863005) were sourced from Bio-Rad (see Key Resources Table for primer and probe sequences). The ddPCR plate was sealed with a Pierceable Foil Heat Seal (Bio-Rad Cat #1814040) and the PX1 PCR Plate Sealer (Bio-Rad Cat #1814000). Plates were transferred to a Bio-Rad Thermalcycler C1000 (Bio-Rad Cat #1851197). The cycling protocol was the following: 45 °C reverse transcription step for 60 min, 95 °C enzyme activation step for 10 min followed by 40 cycles of a two-step cycling protocol (denaturation step of 95 °C for 30 s and annealing/extension step of 58 °C for 1 min), 98 °C enzyme deactivation step for 10 min, and holding at 12 °C for an indefinite period before transfer to the QX200 Droplet Generator. The plates were read with the Bio-Rad QX200 Droplet Generator & Reader

(Cat #1864003) and the RNA concentration per sample was processed using QuantaSoft (Bio-Rad Cat #1864011). Data were exported to Microsoft Excel and Prism (Graphpad) for further statistical analysis.

Most plasmids, including *Nematostella vectensis cacnb2.1* (NveCa$_V$β), *Exaiptasia diaphana* Ca$_V$βs (EdCa$_V$β1, EdCa$_V$β2), *Cyanea capillata* Ca$_V$β (CcCa$_V$β), *Physalia physalis* Ca$_V$β (PpCa$_V$β), *Clytia hemisphaerica* Ca$_V$β (ChCa$_V$β), *Cassiopea xamachana* Ca$_V$β (CxCa$_V$β2), *Nematostella vectensis cacnb2.1* mutant (NveCa$_V$β-N), *R. norvegicus* (rat) *cacnb2a* mutant (rCa$_V$β-N), *Exaiptasia diaphana* Ca$_V$β mutants (EdCa$_V$β1-N, EdCa$_V$β2-N), *Cyanea capillata* Ca$_V$β mutant (CcCa$_V$β-N), *Physalia physalis* Ca$_V$β mutant (PpCa$_V$β-N), *Clytia hemisphaerica* Ca$_V$β mutant (ChCa$_V$β-N), *Cassiopea xamachana* Ca$_V$β mutant (CxCa$_V$β2-N), were synthesized by Genscript (Piscataway, NJ). Sequence alignments were carried out using Clustal Omega. Wild type and Chimeric Ca$_V$β sequences are listed in *Figure 5—source data 1*.

Cnidarian beta sequences were obtained from RNA sequencing or NCBI: *Nematostella vectensis cacnb2.1* (NveCa$_V$β) sequence (*Weir et al., 2020*), *Exaiptasia diaphana* Ca$_V$βs from RNA sequencing and confirmation from NCBI accession number KXJ28099.1 (EdCa$_V$β1) and NCBI accession number XP_020893045.1 (EdCa$_V$β2), *Cyanea capillata* Ca$_V$β (CcCa$_V$β) from NCBI accession number AAB87751.1 (*Jeziorski et al., 1998*), *Physalia physalis* Ca$_V$β (PpCa$_V$β) from NCBI accession number ABD59026 (*Bouchard et al., 2006*), *Clytia hemisphaerica* Ca$_V$β (ChCa$_V$β) from the MARIMBA database (*Leclère et al., 2019*), *Cassiopea xamachana* Ca$_V$β (CxCa$_V$β2) from RNA sequencing as TRINITY_DN5778_c3_g1_i5.p1.

## Transcriptomics

*Exaiptasia* were anesthetized in 15% MgCl$_2$ NSW solution in a dish surrounded by an ice bath for 15 min. Tentacle, body, and acontia tissue were dissected and flash frozen in the presence of liquid nitrogen. *Cassiopea xamachana* were anesthetized in 10% MgCl$_2$ NSW solution in a dish surrounded by an ice bath for 15–20 min. Oral arms, bell, and cassiosome tissues were dissected and flash frozen in the presence of liquid nitrogen. All tissues were stored at –80 °C until RNA extraction, library preparation, and RNA sequencing was performed by Genewiz (Azenta) using a HiSeq (2x150 bp) platform. Reads were examined for base quality distribution, kmer frequencies and adapter contamination by position in the read using fastqc (The Babraham Institute Bioinformatics Group), then where relevant, Rcorrector was used to remove erroneous k-mers (*Song and Florea, 2015*) and the FilterUncorrectablePEfastq python script from the Harvard Informatics group was used to discard read pairs. TrimGalore (The Babraham Institute) was then used to remove adapter contamination in reads and where relevant, Bowtie2 (*Langmead and Salzberg, 2012*) was used to remove reads originating from rRNA and Trinity was used to assemble reference transcriptomes de novo (*Grabherr et al., 2011*). Transdecoder was used to identify open reading frames (Haas, BJ) and Diamond used to annotate the transcriptome (*Buchfink et al., 2015*). Reads were pseudo-aligned and transcript abundance (TPM) was quantified using Kallisto (*Bray et al., 2016*) and our novel transcriptome assemblies as a reference and visualization and alignments were performed with Geneious Prime software and/or Clustal Omega (*Madeira et al., 2022*).

## Electrophysiology

Recordings were carried out at room temperature using a MultiClamp 700B amplifier (Axon Instruments) and digitized using a Digidata 1550B (Axon Instruments) interface and pClamp software (Axon Instruments). Whole-cell recording data were filtered at 1 kHz and sampled at 10 kHz. Ca$_V$ activation data were leak-subtracted online using a p/4 protocol, and all membrane potentials were corrected for liquid junction potentials.

For whole-cell nematocyte recordings, borosilicate glass pipettes were polished to 8–10 MΩ for *Nematostella* and 4–6 MΩ for *Exaiptasia*, respectively. The standard *Nematostella* medium was used as the extracellular solution and contained (in mM): 140 NaCl, 3.3 glucose, 3.3 KCl, 3.3 HEPES, 2 CaCl$_2$, 0.5 MgCl$_2$, pH 7.6, 260-280mOsm. The standard *Exaiptasia* medium was used as extracellular solution and contained (in mM): 430 NaCl, 10 KCl, 10 CaCl$_2$, 50 MgCl$_2$, 10 HEPES, pH 7.6, 800-900mOsm. The intracellular solution for both *Nematostella* and *Exaiptasia* contained (in mM): isolating inward currents (in mM): 500 cesium methanesulfonate, 4 MgCl$_2$, 10 CsEGTA, 10 HEPES, 30 sucrose, pH 7.6, 260-280mOsm for *Nematostella* and 800-900mOsm for *Exaiptasia*. For *Nematostella* nematocyte recordings, voltage-dependent inactivation was measured during a 200ms activating pulse of –20 mV

following a series of 1 s pre-pulses ranging from –110mV to 30 mV, holding at –110 mV. Voltage-gated currents were measured through a series of 200ms voltage pulses in 10 mV increments from –110mV to 70 mV, holding at –110 mV. For *Exaiptasia* nematocyte recordings, voltage-dependent inactivation was measured during a 200ms activating pulse of 0 mV following a series of 1 s pre-pulses ranging from –110mV to 30 mV, holding at –110 mV. For both *Nematostella* and *Exaiptasia,* voltage-gated currents were measured through a series of 200ms steps 200ms voltage pulses in 10 mV increments from –110mV to 70 mV, holding at –110 mV. For $Cd^{2+}$ experiments, 500 µM $Cd^{2+}$ (dissolved in water) was applied locally and voltage-dependent activation was assessed through a single 200ms step to 0 mV from a holding potential of –110 mV.

For whole-cell recordings in HEK293 cells, pipettes were 3–6 MΩ. The standard extracellular solution contained (in mM): 140 NaCl, 5 KCl, 10 HEPES, 2 $CaCl_2$, 2 $MgCl_2$, 10 Glucose, pH 7.4, 300-310mOsm. The intracellular solution contained (in mM): 5 NaCl, 140 cesium methanesulfonate, 1 $MgCl_2$, 10 EGTA, 10 HEPES, 10 sucrose, pH 7.2, 300-310mOsm. For $Ca^{2+}$ currents in heterologously expressed channels, voltage-dependent inactivation was measured in one of two ways: (1) during an activating pulse of 0 mV following a series of 1 s pre-pulses ranging from –110mV to 50 mV and holding potential of –80 mV; or (2) during an activating pulse of 0 mV following a series of 1 s pre-pulses ranging from –110mV to 80 mV and holding potential of –90 mV. Voltage-gated $Ca^{2+}$ currents were measured in response to 200ms voltage pulses in 10 mV increments from –130mV to 80 mV with –110 mV holding potential. Voltage-dependent inactivation was quantified as $I/I_{max}$, with $I_{max}$ occurring at the voltage pulse following a –110 mV prepulse. In some instances, inactivation curves could not be fitted with a Boltzmann equation and were instead fitted with an exponential. G-V relationships were derived from I-V curves by calculating G: $G=I_{CaV}/(V_m-E_{rev})$ and fit with a Boltzmann equation. Data was processed and analyzed in Clampfit (pClamp 11 Software Suite, Molecular Devices) and Microsoft Excel and Prism (GraphPad).

## In situ hybridization (BaseScope)

Adult *Exaiptasia* were paralyzed in anesthetic solution (15% $MgCl_2$), rinsed in PBS, then embedded in Tissue-Tek O.C.T. Compound (Sakura Cat #4583) in cryomolds (Sakura Tissue-Tek Cryomold, Intermediate, Cat #4566) and flash frozen on dry ice and stored at –80 °C. Cryostat sections (18–20 µm) were adhered to Fisherbrand Superfrost Plus Microscope Slides (Fisher Scientific Cat #12-550-15) and flash frozen on dry ice and stored at –80 °C until used for BaseScope. The BaseScope Duplex Detection Reagent Kit (Advanced Cell Diagnostics Cat #323800) and the manufacturer's manual (BaseScope Duplex Detection Reagent User Manual, ACDBio) was followed to hybridize custom probes or positive control (Cat #700101) or negative control probes (Cat #700141) to targets in tissue cryosections and amplify signals. Samples were imaged on an Olympus BX41 Phase Contrast & Darkfield Microscope (Olympus Cat #BX41-PH-B) and images were acquired using the Olympus CellSens software and Olympus DP25 5MP Color Firewire Camera.

## Behavior

Discharge of nematocysts was assessed based on well-established assays (*Gitter et al., 1994*; *Watson and Hessinger, 1994*; *Weir et al., 2020*). For assaying discharge, 5 mm round coverslips were coated with a solution of 25% gelatin (w/v) dissolved in NSW (for *Exaiptasia*) or 1/3 NSW (for *Nematostella*) and allowed to cure 3–4 hr prior to use. Coverslips were presented to the animal's tentacles for 5 s and then immediately imaged at ×20 magnification using a transmitted light source. To assay behavioral responses to prey-derived chemicals, freshly hatched brine shrimp were flash frozen and ground to a powder with a mortar and pestle (Fisherbrand), then filtered through a 0.22 µm syringe filter (VWR Cat #28145–501) and osmolarity adjusted for the specific anemone species. Coverslips were submerged in prey extract for 10 seconds then immediately presented to the animal. Nematocytes visualized on coverslips were only those that embedded in the gelatin after discharge. For experiments using pharmacological agents such as $CdCl_2$, coverslips were submerged in a solution of 1 M (*Exaiptasia*) or 10 mM (*Nematostella*) $Cd^{2+}$ in milli-Q water for 10 s then immediately presented to the animal. After performing the experiments, the animals were given several water changes to remove $Cd^{2+}$. Experiments carried out in the absence of extracellular $Ca^{2+}$ were nominally $Ca^{2+}$ free and did not include use of extracellular chelators. The region of the highest density of discharged nematocytes on the coverslip was imaged at 20 X. Images were acquired with MetaMorph Microscopy Automation and

Image Analysis Software (Molecular Devices) and the number of discharged nematocysts was counted by eye. Images were processed in Fiji (ImageJ) (*Schindelin et al., 2012*). *Exaiptasia* and *Nematostella* tentacles were examined by cutting a small portion of exposed tentacles and sandwiched between glass coverslips and then imaged at 20 X with the MetaMorph software.

## Phylogenetic and genomic analyses

To infer exon boundaries and isoforms, we aligned EdCa$_V$β1 (NCBI accession number LJWW01000015.1) and EdCa$_V$β2 (NCBI accession number XM_021037386.2) to the *Exaiptasia diaphana* reference genome (BioProject PRJNA261862) (*Baumgarten et al., 2015*) using GMAP version 2015-07-23 (*Wu and Watanabe, 2005*). For phylogenetic analyses, we aligned nucleotide sequences with MAFFT v.7 (*Katoh and Standley, 2013*). We used ModelFinder (*Kalyaanamoorthy et al., 2017*) to assess the best model of substitution for phylogenetic inference. We estimated a maximum likelihood gene tree in IQ-TREE v2.0 (*Minh et al., 2020*). Support for clades was calculated using ultrafast bootstrap approximation UFBoot2 (*Hoang et al., 2018*). Percentage of identity for amino acids was calculated in overlapping windows.

## Mathematical model: optimal control theory for the stinging response

### Predatory stinging

To model *Nematostella,* we assume the agent stings for predation. We thus introduce the state of starvation, $s$, that ranges from 0 to 1; at $s = 0$ the agent is least starved and at $s = 1$ the agent is most starved (*Figure 2—figure supplement 1A* top). At each time step, the agent decides to perform an action (sting), $a$, representing the intensity of the attack; $a$ ranges from 0 to 1, and is experimentally compared to the fraction of nematocytes fired in the behavioral assay. Each action has a cost that is proportional to the fraction of nematocysts that are fired, $c(a) = c_0 a$ where $c_0$ is the cost of discharging all nematocysts at once, or cost of full discharge, and we first consider $c_0$ constant. Each stinging event has a probability $p(a)$ of achieving successful predation, where $p(a)$ increases with $a$ (more intense attacks are more costly and more likely to succeed). A successful attack leads to the transition to the next state $s'$ where the agent is more satiated $s \rightarrow s' = s - 1$ whereas a failed attack leads to higher starvation state $s \rightarrow s' = s + 1$. The most starved state is absorbing, which is equivalent to a point of no return. A reward $r(s')$ is assigned to the state of starvation reached upon attack, indicating its desirability ($r(s')$ is a decreasing function of $s'$), and the most starved state entails a starvation penalty $r(1) < 0$. Without loss of generality, all costs and rewards are normalized to the penalty of starvation, hence penalty of starvation is $r(1) = -1$. Our goal is to choose actions that maximize the expected sum of all future net rewards (reward - cost) for each state, which is called the value function. As customary in infinite horizon problems, we ensure convergence of the value function by introducing an effective horizon, i.e. by discounting exponentially rewards that are further in the future with a discount rate $\gamma < 1$.

The optimal value of a state, $V^*(s)$ and the corresponding optimal action $a^*(s)$ are obtained by solving the Bellman Optimality equation (*Bellman, 2003*), with the boundary condition $V(1) = 0$.

$$V^*(s) = \max_a \left( p(a)(r(s-1) - c(a) + \gamma V^*(s-1)) + (1 - p(a))(r(s+1) - c(a) + \gamma V^*(s+1)) \right) \quad (1)$$

$$a^*(s) = \arg\max_a \left( p(a)(r(s-1) - c(a) + \gamma V^*(s-1)) + (1 - p(a))(r(s+1) - c(a) + \gamma V^*(s+1)) \right) \quad (2)$$

### Predatory stinging increases with starvation

We solve *Equations (1) and (2)* numerically with the value iteration algorithm (*Sutton, 2018*) and analytically under the assumption that $a^*(s)$ varies slowly with $s$ (see *Asymptotics for predatory stinging*). The asymptotic solution reproduces well the numerical results (compare lines and full circles in *Figure 2B* **right**, where we used $c_0 = 1$ and $p = p_M(2 - a^2)$ and $p_M = 0.8$ and showcased two different functional forms for $r(s)$, $r(s) = 10 atan (1 - s)$ ; $r(s) = 5 cos \left( \frac{s\pi}{2} \right)$). We also explored how well the asymptotic result can capture the trend of the numerical result by varying the parameters in these three different forms for the reward (*Figure 2—figure supplement 2*). The asymptotic solution shows that the stinging response increases with starvation under broad conditions and not only for specific forms of rewards $r$, transitions $p$ and costs $c$, (i.e. as long as $r(s)$ and $p(a)$ are concave functions and $c(a)$ is convex, see *Asymptotics for predatory stinging*). To exemplify the importance of acting optimally to

save resources, we considered two agents, one acting optimally and one acting randomly i.e. shooting with a number of nematocysts uniformly distributed between $a_{min}$ and $a_{max}$. Both agents start at the same starvation state ($s = 0.9$ in **Figure 2D**) and use on average the same number of nematocysts, but the random agent reaches starvation typically in tens of steps, whereas the optimal agent converges to a steady state (around $s = 0.3$ in the figure) and hardly ever reaches severe starvation. Predatory stinging increases with starvation even when costs increase moderately with starvation; it will eventually decrease with starvation when cost increase is exceedingly steep (see *Asymptotics for predatory stinging – changing cost*).

## Asymptotics for predatory stinging

Short-hand notation: $a^* \equiv a^*(s)$. When $a^* \in (0, 1)$ we obtain it by zeroing the derivative with respect to $a$ in **Equation 1**:

$$-c'(a^*) + p'(a^*)\left[r(s-1) + \gamma V^*(s-1)\right] - p'(a^*)\left[r(s+1) + \gamma V^*(s+1)\right] = 0$$

$$\left[r(s-1) + \gamma V^*(s-1)\right] = r(s+1) + \gamma V^*(s+1) + \frac{c'(a^*)}{p'(a^*)} \tag{3}$$

Plugging **Equation 3** into the Bellman **Equation 1** we obtain:

$$V^*(s) = -c(a^*) + r(s+1) + \gamma V^*(s+1) + p(a^*)\frac{c'(a^*)}{p'(a^*)} \tag{4}$$

From **Equation 3** there is also

$$V^*(s-1) = V^*(s+1) + \frac{1}{\gamma}\left[r(s+1) - r(s-1)\right] + \frac{c'(a^*)}{p'(a^*)} \tag{5}$$

**Equations 4 and 5** are two equations in the four unknowns $V^*(s), V^*(s+1), V^*(s-1)$ and $a^*(s)$. These equations can be solved iteratively by coupling all states and using the boundary conditions on the absorbing state. However, the exact iterative solution is not particularly instructive. Instead, we will make a simplifying assumption that leads to a good approximation that can be used to gather a qualitative understanding of the prediction. Assume that $a^*(s)$ varies slowly with $s$ so that $a^* \equiv a^*(s) \approx a^*(s+1) \approx a^*(s-1)$ (better approximations may be achieved by assuming a first order expansion). Then we obtain a third equation by writing **Equation 4** for the state $\bar{s} = s - 1$

$$V^*(s-1) = -c(a^*) + r(s) + \gamma V^*(s) + p(a^*)\frac{c'(a^*)}{p'(a^*)} \tag{6}$$

We can then repeat the trick to obtain a fourth equation. To this end, we first eliminate $V^*(s-1)$ by combining **Equations 5 and 6**:

$$V^*(s+1) = K(a^*)\frac{\gamma}{1-\gamma^2} - \frac{r(s+1)}{\gamma} + \frac{r(s-1)}{\gamma(1-\gamma^2)} + \frac{r(s)}{1-\gamma^2} \tag{7}$$

$$K(a^*) = -c(a^*)\frac{1+\gamma}{\gamma} + \frac{c'(a^*)}{p'(a^*)}\left(p(a^*)\frac{1+\gamma}{\gamma} - \frac{1}{\gamma^2}\right) \tag{8}$$

Repeating the trick, we can write **Equation 7** for $\bar{s} = s - 1$ and using that $a^*(s-1) = a^*(s)$ we obtain a fourth equation to close the system:

$$V^*(s) = K(a^*)\frac{\gamma}{1-\gamma^2} - \frac{r(s)}{\gamma} + \frac{r(s-2)}{\gamma(1-\gamma^2)} + \frac{r(s-1)}{1-\gamma^2} \tag{9}$$

The system is now closed with the 4 **Equations 4–6; 9** in the 4 unknowns $V^*(s), V^*(s+1), V^*(s-1)$ and $a^*(s)$. We solve for $a^*$ by eliminating $V^*(s)$ from **Equations 4; 9** and plugging the expression for $V^*(s+1)$ from **Equation 7**. After some (tedious) algebra, we obtain that $a^*$ satisfies the simple relation:

$$\frac{p'(a^*)}{c'(a^*)} = \frac{1-\gamma}{-\Delta r(s)} \tag{10}$$

Both the asymptotic solution from *Equation 10* and the numerical solution from value iteration are used in the main text (*Figure 2B* right, symbols and lines respectively). We showcase the robust match between the asymptotic and numerical solutions to *Equation 1* by using a variety of functional forms of the reward function and varying the parameters (*Figure 2—figure supplement 2*). The asymptotics break down if abrupt changes in the rewards and transition rates are assumed, which leads to exceeding slopes in the optimal policy (data not shown). *Equation 12* has a non-trivial solution $0 < a < 1$ when $c' > 0$, $r(s)$ is a decreasing function, and $p(a)$ is an increasing function. If we additionally assume that $r$ is concave, and $p'/c'$ is a decreasing function of $a$ (for example, $p$ is strictly concave and $c$ is convex), then *Equation 12* prescribes that $a$ increases with $s$, as seen graphically in *Figure 2—figure supplement 1A* bottom. Hence independently of the specific functional forms of $c, p$, and $r$, as long as these broad assumptions are valid, optimal stinging for predation entails more intense attacks as starvation increases. For different assumptions of reward function $r(s)$, cost function $c(a)$, and probability $p(a)$, we can easily substitute the specific expressions into *Equation 10* and solve for $a^*$ for every $s$.

## Asymptotics for predatory stinging – changing cost

We applied a cost to predatory stinging that increases with starvation. We found numerically that predatory stinging still increases for moderate increase of $c_0(s)$ with $s$. The result is exemplified in *Figure 2C* using the same functional form for $c_0(s)$ of the defensive stinging (*Figure 2B*, open circles). For a more intense increase of $c_0(s)$ with $s$ predatory stinging eventually decreases with starvation (see *Figure 2—figure supplement 4*, for a comparison with four different cost functions from numerical solutions of *Equations 1; 2*).

These results can be easily understood from our asymptotic solution (10), which appears to still hold when $c = c_0(s)a$ (data not shown -- a formal proof of the asymptotic solution for this case and further consequences for Markov Decision Processes are beyond the scope of the current paper). Indeed, if $c$ increases slightly with $s$, the light blue curve in *Figure 2—figure supplement 3* slightly shifts downward with $s$. If the shift is sufficiently small, its intersection with the green curves still occurs for increasing values of $a$ (dashed line in *Figure 2—figure supplement 3*). However, a dramatic increase of $c$ with $s$ will shift the light-blue curve downward considerably, and the intersection will eventually move backward (dotted line in *Figure 2—figure supplement 3*). In plain words, when the cost of nematocyst discharge for starved animals is dramatically larger than for well-fed animals, the benefits of predation are eventually outweighed by its cost and the animals will sting less with starvation (exemplified in *Figure 2—figure supplement 4*, green and yellow curves). Note that the most extreme increase of cost with starvation is unrealistic as it entails that the cost of stinging is nearly irrelevant when well fed and outweighs the benefits of feeding when starving (see green and yellow cost functions in *Figure 2—figure supplement 4A*). This scenario may become more relevant upon severe starvation, which we do not explore experimentally.

## Defensive stinging

To model *Exaiptasia*, we assume the agent stings for defense, thus the associated Markov process models transitions between the states of safety, which we indicate with *L*, and danger, which we indicate with *D* (*Figure 2—figure supplement 1A* top). The state of starvation is not affected by stinging and instead is dictated by a separate process that relies on symbionts and which we do not model. Similar to the previous model, the agent chooses an action, $a$, representing the intensity of the attack. Each attack has a likelihood to succeed $p(a)$ and an associated cost $c(a) = c_0 a$ where $c_0$ is the cost of full discharge of all nematocysts at once. A successful attack allows the animal to remain in state *L* and receive a unit reward; a failed attack leads to state *D* and penalty -1. *F* is an absorbing state hence $V^*(F) = 0$. The optimal value and action in state *L* follow:

$$V^*(L) = \max_a \left( p(a)(-c_0 a + \gamma V^*(L) + 1) + (1 - p(a))(-1 - c_0 a) \right)$$

$$a^*(L) = \arg\max_a \left( p(a)(-c_0 a + \gamma V^*(L) + 1) + (1 - p(a))(-1 - c_0 a) \right)$$

(11)

## Analytic solution for defensive stinging

Zeroing the derivative of the argument on the r.h.s. of *Equation 11* leads to

$$-c_0 + p'(a^*)\left(\gamma V^*(L) + 2\right) = 0$$

$$V^*(L) = -c_0 a^* + p(a^*)\left(\gamma V^*(L) + 1\right) + \left(1 - p(a^*)\right)$$

Here, $p(a)$ is the probability of success of action $a$ and it ranges from $p(0) = 0$ to $p(1) = p_M < 1$. Combining these equations we obtain an implicit algebraic equation for $a^*$ :

$$\left(\gamma p(a^*) - 1\right)\left(\frac{c'(a^*)}{p'(a^*)} - 1\right) = \gamma c(a^*) + \gamma\left(1 - p(a^*)\right) - 1 \tag{12}$$

Assuming the specific form $p = p_M a\left(2 - a\right)$ in **Equation 12** leads to the constant optimal action: $a^* = K - \sqrt{K^2 - A}$ , where $K = \frac{2-\gamma}{c_0\gamma}$ and $A = \frac{-1}{\gamma p_M} + 2K$. If $c_0$ is constant, there is a non-trivial optimal action as long as $c_0 < 2p_M\left(2 - \gamma\right)$ and clearly the optimal action does not depend on starvation (constant solution in **Figure 2E** right, with $c_0 = 1$; $\gamma = 0.99$ and $p_M = 0.8$).

### Defensive stinging – increasing cost

Stinging predators does not improve nutritional state, thus transitions among different starvation states are not modelled for defensive stinging, but instead rely heavily on symbionts. However, to capture subtle effects of starvation on defensive stinging, we note that the cost of discharging nematocysts may still depend parametrically on whatever state of starvation the agent happens to be in. In this case, $c_0 = c_0(s)$ and under the assumption that the cost increases with starvation, we find that optimal defensive stinging always decreases with starvation, for any functional form of $p$. Indeed, **Equation 12** with $c_s(a) = c(s, a)$ simply reads:

$$\left(\gamma p(a^*) - 1\right)\left(\frac{\partial_a c_s(a^*)}{p'(a^*)} - 1\right) = \gamma c_s(a^*) + \gamma\left(1 - p(a^*)\right) - 1 \tag{13}$$

where $\partial_a c$ is now a partial derivative with respect to $a$. For $c = c_0(s)a$ and $p = p_M a\left(2 - a\right)$ , the solution is $a^*(s) = K(s) - \sqrt{K(s)^2 - C(s)}$ where $K(s) = \varepsilon/c_0(s)$ ; $\varepsilon = \left(2 - \gamma\right)/\gamma$ , $C(s) = -1/\left(p_M\gamma\right) + 2\varepsilon/c_0(s)$ and it exists for $c_0(s) < 2p_M\left(2 - \gamma\right)$ . Outside these boundaries, the solution is either $a^*=0$ or $a^*=1$ and it is obtained by comparing $V(L)$ for these two choices of action and choosing the one that maximizes $V$. This solution is decreasing as can be easily demonstrated by deriving with respect to $s$. We used this solution for $a^*(s)$ to match the experimental data and obtained a fit for the cost function shown in **Figure 2A** (empty circles).

While we discussed a specific case for the choices of $p$ and $a$ above, optimal defensive stinging decreases with $s$ in much more general conditions. Indeed, $a$ decreases or remains constant with $s$ using the same broad classes of functions discussed for predatory stinging ($p(a)$ is concave and $c$ is convex in $a$; either $p$ or $c$ can be linear in $a$, but not both) and assuming additionally that $dc_0/ds \geq 0$ i.e. that the cost does not decrease with starvation. Rearranging **Equation 13** we note that $a$ is defined by the point where:

$$\frac{1 - \gamma p(a^*)}{p'(a^*)}\partial_a c_s(a^*) + \gamma c_s(a^*) = 2 - \gamma \tag{14}$$

The l.h.s. of **Equation 14** is an increasing function of $a$ as seen by deriving with respect to $a$ and using the assumptions: $p < 1$; $p'' \leq 0$; $c' \geq 0$ and $c'' \geq 0$. Because $c_s(a)$ increases with $s$, the intersection of the l.h.s. with the constant value $2 - \gamma$ occurs at lower and lower values of $a$ as $s$ increases (see graphical representation **Figure 2—figure supplement 1B** bottom). Thus under the same broad assumptions for the functional forms of $c$ and $p$, stinging for predation increases with starvation, whereas stinging for defense remains constant or decreases with starvation.

## Statistical analysis

Data were analyzed with Clampfit (Axon Instruments), Prism (GraphPad), or QuantaSoft (BioRad Laboratories) and are represented as mean ± sem. $n$ represents independent experiments for the number of cells/patches or behavioral trials. Data were considered significant if $P<0.05$ using paired or unpaired two-tailed Student's t-tests or one- or two-way ANOVAs. All significance tests were justified considering the experimental design and we assumed normal distribution and variance, as is common

for similar experiments. Sample sizes were chosen based on the number of independent experiments required for statistical significance and technical feasibility.

## Acknowledgements

We thank B Walsh and P Kilian for assistance with animal husbandry, A Whipple and D Loftus with guidance for ddPCR experiments, A Grearson for illustrations and photographs, K Koenig and M Martindale, and the Marine Biological Laboratory for providing animals. We also thank the Harvard Center for Biological Imaging (RRID:SCR_018673), Histology Core at the Harvard Department of Stem Cell and Regenerative Biology, and The Bauer Core Facility at Harvard University for infrastructure and experimental support. This research was supported by grants to NWB from the New York Stem Cell Foundation, Searle Scholars Program, and the NIH (R35GM142697), fellowships to LH from NSF Graduate Research Fellowship Program and Physics of Living Systems (PoLS) Graduate Fellowship and the Simmons Award at the Harvard Center for Biological Imaging, grants to AS from the European Research Council (ERC) under the European Union's Horizon 2020 research and innovation programme (grant agreement No 101002724 RIDING), the Air Force Office of Scientific Research under award number FA8655-20-1-7028, and the National Institutes of Health (NIH) under award number R01DC018789.

## Additional information

### Competing interests

Agnese Seminara: Reviewing editor, eLife. The other authors declare that no competing interests exist.

### Funding

| Funder | Grant reference number | Author |
| --- | --- | --- |
| National Institutes of Health | R35GM142697 | Nicholas W Bellono |
| European Research Council | 101002724 RIDING | Agnese Seminara |
| Air Force Office of Scientific Research | FA8655-20-1-7028 | Agnese Seminara |
| National Institutes of Health | R01DC018789 | Agnese Seminara |
| New York Stem Cell Foundation | | Nicholas W Bellono |
| Searle Scholars Program | | Nicholas W Bellono |
| National Science Foundation | Graduate Research Fellowship Program | Lily S He |
| National Science Foundation | Physics of Living Systems (PoLS) Graduate Fellowship | Lily S He |
| Harvard University | Simmons Award at the Harvard Center for Biological Imaging | Lily S He |

The funders had no role in study design, data collection and interpretation, or the decision to submit the work for publication.

### Author contributions

Lily S He, Conceptualization, Data curation, Formal analysis, Methodology, Writing – original draft, Writing – review and editing; Yujia Qi, Keiko Weir, Conceptualization, Data curation, Formal analysis, Writing – review and editing; Corey AH Allard, Conceptualization, Data curation, Methodology,

Writing – review and editing; Wendy A Valencia-Montoya, Stephanie P Krueger, Data curation, Formal analysis, Writing – review and editing; Agnese Seminara, Conceptualization, Formal analysis, Supervision, Funding acquisition, Writing – original draft, Project administration, Writing – review and editing; Nicholas W Bellono, Conceptualization, Supervision, Funding acquisition, Writing – original draft, Project administration, Writing – review and editing

### Author ORCIDs
Lily S He http://orcid.org/0000-0003-2987-3393
Agnese Seminara https://orcid.org/0000-0001-5633-8180
Nicholas W Bellono http://orcid.org/0000-0002-0829-9436

Reviewer #1 (Public Review): https://doi.org/10.7554/eLife.88900.3.sa1
Reviewer #2 (Public Review): https://doi.org/10.7554/eLife.88900.3.sa2
Reviewer #3 (Public Review): https://doi.org/10.7554/eLife.88900.3.sa3
Author Response https://doi.org/10.7554/eLife.88900.3.sa4

## Additional files

### Supplementary files
• MDAR checklist

### Data availability
Deep sequencing data are available via the Sequence Read Archive (SRA) repository under the BioProject accession code PRJNA945904. All plasmids are available upon request. Further requests for resources and reagents should be directed to and will be fulfilled by the corresponding author, NWB (nbellono@harvard.edu). The Matlab code to obtain the optimal predicted stinging according to our Markov Decision Process is available from Zenodo.

The following datasets were generated:

| Author(s) | Year | Dataset title | Dataset URL | Database and Identifier |
|---|---|---|---|---|
| Valencia-Montoya WA, He LS, Bellono NW | 2023 | Molecular tuning of sea anemone stinging RNA-seq | https://www.ncbi.nlm.nih.gov/bioproject/?term=PRJNA945904 | NCBI BioProject, PRJNA945904 |
| Qi Y | 2023 | MDP-Anemone Code | https://doi.org/10.5281/zenodo.8177567 | Zenodo, 10.5281/zenodo.8177567 |

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
