## [Editor Report · eLife assessment]

This is an **important** paper that links distinctive stinging behavior of two related anemones occupying different ecological niches to varying inactivation properties of voltage-gated calcium channels conferred by auxiliary Cavβ subunits. Further **convincing** evidence is provided that these differences are mediated by alternative splicing of Cavβ subunit of the calcium channel. The study will be of interest to scientists studying Ca2+ signaling, ion channel biophysicists, and marine biologists.

---

## [Referee Report · Reviewer #1 (Public Review)]

This manuscript by He et al. explores the molecular basis of the different stinging behaviors of two related anemones. The freshwater Nematostella which only stings when a food stimulus is presented with mechanical stimulation and the saltwater Exaiptasia which stings in response to mechanical stimuli. The authors had previously shown that Nematostella stinging is calcium-dependent and mediated by a voltage-gated calcium channel (VGCC) with very pronounced voltage-dependent inactivation, which gets removed upon hyperpolarization produced by taste receptors.

In this manuscript, they show that Exaiptacia and Nematostella differing stinging behavior is near optimal, according to their ecological niche, and conforms to predictions from a Markov decision model.

It is also shown that Exaiptacia stinging is also calcium-dependent, but the calcium channel responsible is much less inactivated at resting potential and can readily induce nematocyte discharge only in the presence of mechanical stimulation. To this end, the authors record calcium currents from Exaipacia nematocysts and discover that the VGCCs in this anemone are not strongly inactivated and thus are easily activated by mechanical stimuli-induced depolarization accounting for the different stinging behavior between species. The authors further explore the role of the auxiliary beta subunit in the modulation of VGCC inactivation and show that different n-terminal splice variants in Exaiptacia produce strong and weak voltage-dependent inactivation.

The manuscript is clear and well-written and the conclusions are in general supported by the experiments and analysis. The findings are very relevant to increase our understanding of the molecular basis of non-neural behavior and its evolutionary basis. This manuscript should be of general interest to biologists as well as to more specialized fields such as ion channel biophysics and physiology.

---

## [Referee Report · Reviewer #2 (Public Review)]

This manuscript links the distinctive stinging behavior of sea anemones in different ecological niches to varying inactivation properties of voltage-gated calcium channels that are conferred by the identity of auxiliary Cavbeta subunits. Previous work from the Bellono lab established that the burrowing anemone, *Nematostella vectensis*, expresses a CaV channel that is strongly inactivated at rest which requires a simultaneous delivery of prey extract and touch to elicit a stinging response, reflecting a precise stinging control adapted for predation. They show here that by contrast, the anemone Exaiptasia diaphana which inhabits exposed environments, indiscriminately stings for defense even in the absence of prey chemicals, and that this is enabled by the expression of a CaVbeta splice variant that confers weak inactivation. They further use the heterologous expression of CaV channels with wild type and chimeric anemone Cavbeta subunits to infer that the variable N-termini are important determinants of Cav channel inactivation properties.

---

## [Referee Report · Reviewer #3 (Public Review)]

Summary:

The present article attempts to answer both the ultimate question of why different stinging behaviours have evolved in Cnidiarians with different ecological niches and shed light on the proximate question of which electro-physiological mechanisms underlie these distinct behaviours.

Account of major methods and results:

In the first part of the paper, the authors try to answer the ultimate question of why distinct dependencies of the sting response on internal starvation levels have evolved. The premise of the article that Exaiptasia's nematocyte discharge is independent of the presence of prey (Artemia nauplii) as compared to Nematostella's significant dependence of the discharge on the presence of actual prey, is shown to be a robust phenomenon justified by the data in Figure 1.

The hypothesis that defensive vs. predatory stinging leads to different nematocyte discharge behaviours is analysed in mathematical models based on the suitable framework of optimal control/decision theory. By assuming functional relations between the:

1. cost of a full nematocyte discharge and the starvation level.

2. probability of successful predation/avoidance on the discharge level.

3. desirability/reward of the reached nutritional state.

Based on these assumptions of environmental and internal influences, the optimal choice of attack intensity is calculated using Bellman's equation for this problem. The model predictions are validated using counted nematocytes on a coverslip. The scaling of normalised nematocyte discharge numbers with scaled starvation time is qualitatively comparable to what is predicted from the models. The abundance of nematocytes in the tentacles was, on the other hand, independent of the starvation state of the animals.

Next, the authors turn to investigate the proximate cause of the differential stinging behaviour. The authors have previously reported convincing evidence that a strongly inactivating Cav2.1 channel ortholog (nCav) is used by Nematostella to prevent stinging in the absence of prey (Weir et al. 2020). This inactivation is released by hyperpolarising sensory inputs signalling the presence of prey. In this article, it is clearly shown by blocking respective currents that Exaiptasia, too, relies on extracellular Ca2+ influx to initiate stinging. Patch clamp data of the involved currents is provided in support. However, the authors find that in addition to the nCav with a low-inactivation threshold, Exaiptasia has a splice variant with a higher inactivation threshold expressed (Figure 3D).

The authors hypothesise that it is this high-threshold nCav channel population that amplifies any voltage depolarisation to release a sting irrespective of the presence of prey signals. They found that the β subunit that is responsible for Nematostella's unusually low inactivation threshold exists in Exaiptasia as two alternative splice isoforms. These N-terminus variants also showed the greatest variation in a phylogenetic comparison (Figure 5), rendering it a candidate target for mutations causing variation in stinging responses.

Appraisal of methodology in support of the conclusions:

The authors base their inference on a normative model that yields quantitative predictions which is an exciting and challenging approach. The authors take care in stating the model assumptions as well as showing that the data indeed does not contradict their model predictions. The interesting comparative nature of the modelling part of the study is complicated by slightly different cost assumptions for the two scenarios. Hence, Figure 2 needs to be carefully digested by readers.

It would be even more prudent to analyse the same set of cost-of-discharge vs. starvation scenarios for both species. Specifically, for Nematostella the complete cost-of-discharge vs starvation-state curves as for Exaiptasia (Figure 2E, example 2-4) could be used. It is likely that the differential effect size of Nematostella and Exaiptasia behaviour is the strongest if only the flat cost-of-discharge vs starvation is used (Figure 2A) for Nematostella. But as a worst-case comparison the other curves, where the cost to the animal scales with starvation would be a good comparison. This could help the reader to understand when the different prediction of Nematostella's behaviour breaks down. In addition, this minor change could shed light on broader topics like common trade-offs in pursuit predation.

The qualitatively similar scaling of the model-derived relation between starvation and sting intensity with the counted nematocytes for different feeding pauses is evidence that feeding has indeed been optimised for the two distinct ecological niches.

To prove that Exaiptasia uses a similar Ca2+ channel ortholog as well as a different splice variant, the authors employed both clean electrophysiological characterisation (Figure 3) as well as transcriptomics data (Figure 4S1).

To strengthen the authors' hypothesis that variation in the N-termini leads to changes in Ca2+ channel inactivation and hence altered stinging, the response sequence variability of 6 Cnidaria was analysed.

Additional context:

Although, the present article focuses on nematocytes alone, currently, there has been a refocus in neurobiology on the nervous systems of more basal metazoans, which received much attention already in the works of Romanes (1885). In part, this is driven by the goal to understand the early evolution of nervous systems. Cnidarians and Ctenophors are exciting model organisms in this venture. This will hopefully be accompanied by more comparative studies like the present one. Some of the recent literature also uses computational models to understand mechanisms of motor behaviour using full-body simulations (Pallasdies et al. 2019; Wang et al. 2023), which can be thought of as complementary to the normative modelling provided by the authors.

Comparative studies of recent Cnidarians, such as the present article, can shed light on speculative ideas on the origin of nervous systems (Jékely, Keijzer, and Godfrey-Smith 2015). During a time (the Ediacarium/Cambrium transition) that has seen the genesis of complex trophic foodwebs with preditor-prey interaction, symbioses, but also an increase of body sizes and shapes, multiple ultimate causes can be envisioned that drove the increase in behavioural complexity. The authors show that not all of it needs to be implemented in dedicated nerve cells.

References:

Jékely, Gáspár, Fred Keijzer, and Peter Godfrey-Smith. 2015. "An Option Space for Early Neural Evolution." Philosophical Transactions of the Royal Society B: Biological Sciences 370 (December): 20150181. https://doi.org/10.1098/rstb.2015.0181.

Pallasdies, Fabian, Sven Goedeke, Wilhelm Braun, and Raoul-Martin Memmesheimer. 2019. "From Single Neurons to Behavior in the Jellyfish Aurelia Aurita." eLife 8 (December). https://doi.org/10.7554/elife.50084.

Romanes, G. J. 1885. Jelly-Fish, Star-Fish and Sea-Urchins: Being a Research on Primitive Nervous Systems. Appleton.

Wang, Hengji, Joshua Swore, Shashank Sharma, John R. Szymanski, Rafael Yuste, Thomas L. Daniel, Michael Regnier, Martha M. Bosma, and Adrienne L. Fairhall. 2023. "A Complete Biomechanical Model of hydra Contractile Behaviors, from Neural Drive to Muscle to Movement." Proceedings of the National Academy of Sciences 120 (March). https://doi.org/10.1073/pnas.2210439120.

Weir, Keiko, Christophe Dupre, Lena van Giesen, Amy S-Y Lee, and Nicholas W Bellono. 2020. "A Molecular Filter for the Cnidarian Stinging Response." eLife 9 (May). https://doi.org/10.7554/elife.57578.

---

## [Author Response]

The following is the authors’ response to the original reviews.

**Reviewer #1 (Public Review):**
This manuscript by He et al. explores the molecular basis of the different stinging behaviors of two related anemones. The freshwater Nematostella which only stings when a food stimulus is presented with mechanical stimulation and the saltwater Exaiptasia which stings in response to mechanical stimuli. The authors had previously shown that Nematostella stinging is calcium-dependent and mediated by a voltage-gated calcium channel (VGCC) with very pronounced voltage-dependent inactivation, which gets removed upon hyperpolarization produced by taste receptors.In this manuscript, they show that Exaiptacia and Nematostella differing stinging behavior is near optimal, according to their ecological niche, and conforms to predictions from a Markov decision model.It is also shown that Exaiptacia stinging is also calcium-dependent, but the calcium channel responsible is much less inactivated at resting potential and can readily induce nematocyte discharge only in the presence of mechanical stimulation. To this end, the authors record calcium currents from Exaipacia nematocysts and discover that the VGCCs in this anemone are not strongly inactivated and thus are easily activated by mechanical stimuli-induced depolarization accounting for the different stinging behavior between species. The authors further explore the role of the auxiliary beta subunit in the modulation of VGCC inactivation and show that different n-terminal splice variants in Exaiptacia produce strong and weak voltage-dependent inactivation.The manuscript is clear and well-written and the conclusions are in general supported by the experiments and analysis. The findings are very relevant to increase our understanding of the molecular basis of non-neural behavior and its evolutionary basis. This manuscript should be of general interest to biologists as well as to more specialized fields such as ion channel biophysics and physiology.Some findings need to be clarified and perhaps additional experiments performed.1. The authors identify by sequencing that the Exaiptacia Cav is a P-type channel (cacna1a). However, the biophysical properties of the nematocyte channel are different from mammalian P-type channels. The cnidarian channel inactivation is exceedingly rapid and activation happens at relatively low voltages. These substantial differences should be mentioned and commented on.

First, we thank Reviewer 1 for thoughtful and detail-oriented comments, as well as their shared appreciation for the molecular basis of unique behaviors. Indeed, Nematostella and rat CaV channels exhibit striking differences in inactivation (both fast and steady-state). We previously described this in Weir et al., 2020 and added additonal text to ensure that this result is clear.

1. The currents from Nematostella in Figure 3d seem to be poorly voltage-clamped. Poor voltage-clamp is also evident in the sudden increase of conductance in Figure 3C and might contribute to incorrect estimation of voltage dependence of activation and if present in inactivation experiments, also to incorrect estimation of the inactivation voltage range. This problem should be reassessed with new data.

Because it is necessary to use small-tipped pipettes to get recordings from small and technically challenging nematocytes, there is imperfect voltage clamp that is evident in the steep activation curves. This issue should have little effect on the inactivation curves determined with 1s pre-pulses because poor voltage control occurs transiently at the beginning of the pre-pulse. In our case, current is measured in response to a brief maximally activating pulse followed by a nearly 1s period. Thus, error should be minimal in inactivation curves if the test pulse is a maximally activating voltage. We ensured that these protocols are clearly described in the Methods to address this issue. In addition, we are confident in the described inactivation values because they are generally consistent with channel properties measured in a heterologous expression system in which we do not have this problem and see the same differences in inactivation (also see Weir et al., 2020).

1. While co-expression of the mouse Cav channel with the beta1 isoform from Exaiptacia indeed shifts inactivation to more negative voltages, it does not recapitulate the phenotype of the more inactivated Ca-currents in nematocytes (compare Figures 4d and 5d). It should be explained if this might be due to the use of a mammalian alpha subunit. Related to this, did the authors clone the alpha subunit from Exaiptacia? Using this to characterize the effect of beta subunits on inactivation might be more accurate.

While the cnidarian CaVβ subunits indeed shift inactivation consistent with native properties, we agree that using the Exaiptasia alpha subunit would be more accurate. We were unable to successfully clone and heterologously express this subunit, however, we did express all subunits from Nematostella and made chimeric channels in which alpha, alpha2d, or CaVβ were swapped between Nematostella and mammalian channels. These experiments demonstrated the requirement and sufficiency of the CaVβ subunit in altering inactivation (Weir et al., 2020). Furthermore, we were able to express CaVβ subunits from a variety of other cnidarians, all of which affected inactivation properties. Thus, we are confident in the conclusion that CaVβ subunits are major contributors to molecular tuning of cnidarian CaV channels. Future studies aim to incorporate describing properties of the alpha subunit from Exaiptasia and other cnidarians.

1. The in situ shown in Figure 4b are difficult to follow for a non-expert in cnidarian anatomy. Some guidance should be provided to understand the structures. Also, for the left panels, is the larger panel the two-channel image? If so, blue would indicate co-localization of the two isoforms and there seems to be a red mark in the same nematocyte.

We thank the reviewer for this important comment and have modified the figure to enhance visual guidance. We more clearly highlighted the nematocyte in the single and two-channel images and selected the clearest representative images. For additional reference, previous studies beautifully illustrate the unusual morphology of nematocytes, including the relative localization of the nematocyst and nucleus in the context of cnidarian tissues (Babonis and Martindale, 2017).

**Reviewer #2 (Public Review):**
This manuscript links the distinctive stinging behavior of sea anemones in different ecological niches to varying inactivation properties of voltage-gated calcium channels that are conferred by the identity of auxiliary Cavbeta subunits. Previous work from the Bellono lab established that the burrowing anemone, *Nematostella vectensis*, expresses a CaV channel that is strongly inactivated at rest which requires a simultaneous delivery of prey extract and touch to elicit a stinging response, reflecting a precise stinging control adapted for predation. They show here that by contrast, the anemone Exaiptasia diaphana which inhabits exposed environments, indiscriminately stings for defense even in the absence of prey chemicals, and that this is enabled by the expression of a CaVbeta splice variant that confers weak inactivation. They further use the heterologous expression of CaV channels with wild type and chimeric anemone Cavbeta subunits to infer that the variable N-termini are important determinants of Cav channel inactivation properties.1. The authors found that Exaiptasia nematocytes could be characterized by two distinct inactivation phenotypes: (1) nematocytes with low-voltage threshold inactivation similar to that of Nematostella (Vi1/2 = ~ -85mV); and (2) a distinct population with weak, high-voltage threshold inactivation (Vi1/2 = ~ -48mV). What were the relative fractions of low-voltage and high-voltage nematocytes? Do the low-voltage Exaiptasia nematocytes behave similarly to Nematostella nematocytes with respect to requiring both prey extract and touch to discharge?

We thank Reviewer 2 for thoughtful comments and questions. Nematocyte patch clamp is technically challenging due to small size, large nematocyst, and, notably, the explosive discharge involved in stinging! Therefore, we only patch clamped a small number of cells. Despite this limitation, we were able to observe two distinct nematocyte populations based on physiological properties. Yet, we did not observe a correlation with morphology and cannot make broad comments on relative fractions. Because morphology was generally similar and Exaiptasia nematocytes discharge even from touch alone, it remains unclear whether the low-voltage population behaves similarly to Nematostella nematocytes that only discharge in response to chemicals and touch. Future in vivo approaches could be used to address this question.

1. The authors state in Fig 3 legend and in the results that Exaiptasia nematocyte voltage-gated Ca2+ currents have weak inactivation compared with Nematostella. This description is imprecise and inaccurate. Figure 3 in fact shows that Exaiptasia nematocyte voltage-gated Ca2+ currents display a faster rate of inactivation compared to Nematostella Ca2+ currents. A sub-population of Exaiptasia nematocytes does display less resting state (or steady-state) inactivation compared to Nematostella Ca2+ currents. The authors need to be more accurate and qualify what type of inactivation property they are talking about.'

We thank Reviewer 2 for this attention to detail and have defined this phrasing early in the text.

1. In a similar vein, the authors need to be more accurate when referring to 'rat beta' used in heterologous expression experiments. It should be made explicit throughout the manuscript that the rat beta isoform used is rat beta2a. Among the distinct beta isoforms, beta2a is unique in being palmitoylated at the N-terminus which confers a characteristic slow rate of inactivation and a right-shifted voltage-dependence of steady-state inactivation consistent with the data shown in Fig. 4D. Almost all other rat beta isoforms do not have these properties.

We used the rat CaVβ2a for comparison because it shares the highest homology with Nematostella CaVβ (Weir et al., 2020). We have now more clearly defined the rat subunit in the text and legends.

1. The profiling of the impact of different Cnidarian Cavbeta subunits on reconstituted Ca2+ channel current waveforms is nice (Fig 5 and Fig 5S1). The N-terminus sequence of EdCaVβ2 is different from palmitoylated rat beta2a, though both have similar properties in showing slow inactivation and a right-shifted voltage-dependence of steady-state inactivation. Does EdCaVβ2 target autonomously the plasma membrane when expressed in cells? If so, this would reconcile with what was previously known and provide a rational explanation for the observed functional impact of the distinct Cavbetas.

As far as we understand the question, our data support that Exaiptasia CaVβ2 targets the plasma membrane for a number of reasons: (1) Expressing Exaiptasia CaVβ2 produces consistent properties in comparison with other CaVβs, suggesting a homogenous population of channel complexes; (2) Distinct cnidarian-Exaiptasia CaVβ2 chimeras produce distinct and internally consistent properties; and (3) Expressing P/Q-type CaV alpha + alpha2d subunits without CaVβ in cell lines does not produce robust measurable voltage-gated currents. We further tested this in our case and found the same result: at an equivalent maximally activating step using the same protocol, we measured 458.68 ± 179.88pA average current amplitude for +Exaiptasia CaVβ2 (n = 6) and 43.03 ± 17.64pA average current amplitude for -CaVβ2 (n = 4).

**Reviewer #3 (Public Review):**
Summary:The present article attempts to answer both the ultimate question of why different stinging behaviours have evolved in Cnidiarians with different ecological niches and shed light on the proximate question of which electro-physiological mechanisms underlie these distinct behaviours.Account of major methods and results:In the first part of the paper, the authors try to answer the ultimate question of why distinct dependencies of the sting response on internal starvation levels have evolved. The premise of the article that Exaiptasia's nematocyte discharge is independent of the presence of prey (Artemia nauplii) as compared to Nematostella's significant dependence of the discharge on the presence of actual prey, is shown be a robust phenomenon justified by the data in Figure 1.The hypothesis that defensive vs. predatory stinging leads to different nematocyte discharge behaviours is analysed in mathematical models based on the suitable framework of optimal control/decision theory. By assuming functional relations between the:1. cost of a full nematocyte discharge and the starvation level.1. probability of successful predation/avoidance on the discharge level.1. desirability/reward of the reached nutritional state.Based on these assumptions of environmental and internal influences, the optimal choice of attack intensity is calculated using Bellman's equation for this problem. The model predictions are validated using counted nematocytes on a coverslip. The scaling of normalised nematocyte discharge numbers with scaled starvation time is qualitatively comparable to what is predicted from the models. The abundance of nematocytes in the tentacles was, on the other hand, independent of the starvation state of the animals.Next, the authors turn to investigate the proximate cause of the differential stinging behaviour. The authors have previously reported convincing evidence that a strongly inactivating Cav2.1 channel ortholog (nCav) is used by Nematostella to prevent stinging in the absence of prey (Weir et al. 2020). This inactivation is released by hyperpolarising sensory inputs signalling the presence of prey. In this article, it is clearly shown by blocking respective currents that Exaiptasia, too, relies on extracellular Ca2+ influx to initiate stinging. Patch clamp data of the involved currents is provided in support. However, the authors find that in addition to the nCav with a low-inactivation threshold, Exaiptasia has a splice variant with a higher inactivation threshold expressed (Figure 3D).The authors hypothesise that it is this high-threshold nCav channel population that amplifies any voltage depolarisation to release a sting irrespective of the presence of prey signals. They found that the β subunit that is responsible for Nematostella's unusually low inactivation threshold exists in Exaiptasia as two alternative splice isoforms. These N-terminus variants also showed the greatest variation in a phylogenetic comparison (Figure 5), rendering it a candidate target for mutations causing variation in stinging responses.Appraisal of methodology in support of the conclusions:The authors base their inference on a normative model that yields quantitative predictions which is an exciting and challenging approach. The authors take care in stating the model assumptions as well as showing that the data indeed does not contradict their model predictions. The interesting comparative nature of the modelling part of the study is complicated by slightly different cost assumptions for the two scenarios. Hence, Figure 2 needs to be carefully digested by readers.

We thank the reviewer for their careful revision of our work and excellent comments. We simplified Figure 2 considerably to make it easier to digest. We now compare the stinging response for predation vs defense under the same exact definition of cost per nematocyte for both models. You can find examples 1 and 2 in Figure 2 and examples 3 and 4 in Supplementary Figure 3 (see response below).

It would be even more prudent to analyse the same set of cost-of-discharge vs. starvation scenarios for both species. Specifically, for Nematostella the complete cost-of-discharge vs starvation-state curves as for Exaiptasia (Figure 2E, example 2-4) could be used. It is likely that the differential effect size of Nematostella and Exaiptasia behaviour is the strongest if only the flat cost-of-discharge vs starvation is used (Figure 2A) for Nematostella. But as a worst-case comparison the other curves, where the cost to the animal scales with starvation would be a good comparison. This could help the reader to understand when the different prediction of Nematostella's behaviour breaks down. In addition, this minor change could shed light on broader topics like common trade-offs in pursuit predation.

The results hold even when the cost increases moderately with starvation: Figure 2 now shows results with the same cost for predatory and defensive stinging (cost defined in Figure 2A, former examples 1 and 4). Predatory stinging robustly increases with starvation and defensive stinging remains constant or decreases. Interestingly, the fit between theory and data for both anemones improves by using the increasing cost (open circles in Figure 2E right). For other choices of increasing cost functions, defensive stinging will always decrease, and even more so if the cost increases dramatically (like for the former Examples 2 and 3). In contrast, predatory stinging will switch behavior if the cost increases too much with starvation (results with former Examples 2 and 3, now in Supplementary Figure 3 and theoretical arguments in Supplementary Information). Note however that these assumptions are less realistic because they necessitate that the cost of stinging for well-fed animals is negligible with respect to the cost for starved animals. A formal proof of the asymptotic solution for predatory stinging with varying cost is beyond the scope of this work and is subject of ongoing work where we consider implications for Markov Decision Processes in continuous space state.

The qualitatively similar scaling of the model-derived relation between starvation and sting intensity with the counted nematocytes for different feeding pauses is evidence that feeding has indeed been optimised for the two distinct ecological niches.To prove that Exaiptasia uses a similar Ca2+ channel ortholog as well as a different splice variant, the authors employed both clean electrophysiological characterisaiton (Figure 3) as well as transcriptomics data (Figure 4S1).To strengthen the authors' hypothesis that variation in the N-termini leads to changes in Ca2+ channel inactivation and hence altered stinging, the response sequence variability of 6 Cnidaria was analysed.Additional context:Although, the present article focuses on nematocytes alone, currently, there has been a refocus in neurobiology on the nervous systems of more basal metazoans, which received much attention already in the works of Romanes (1885). In part, this is driven by the goal to understand the early evolution of nervous systems. Cnidarians and Ctenophors are exciting model organisms in this venture. This will hopefully be accompanied by more comparative studies like the present one. Some of the recent literature also uses computational models to understand mechanisms of motor behaviour using full-body simulations (Pallasdies et al. 2019; Wang et al. 2023), which can be thought of as complementary to the normative modelling provided by the authors.Comparative studies of recent Cnidarians, such as the present article, can shed light on speculative ideas on the origin of nervous systems (Jékely, Keijzer, and Godfrey-Smith 2015). During a time (the Ediacarium/Cambrium transition) that has seen the genesis of complex trophic foodwebs with preditor-prey interaction, symbioses, but also an increase of body sizes and shapes, multiple ultimate causes can be envisioned that drove the increase in behavioural complexity. The authors show that not all of it needs to be implemented in dedicated nerve cells.References:Jékely, Gáspár, Fred Keijzer, and Peter Godfrey-Smith. 2015. "An Option Space for Early Neural Evolution." Philosophical Transactions of the Royal Society B: Biological Sciences 370 (December): 20150181. https://doi.org/10.1098/rstb.2015.0181.Pallasdies, Fabian, Sven Goedeke, Wilhelm Braun, and Raoul-Martin Memmesheimer. 2019. "From Single Neurons to Behavior in the Jellyfish Aurelia Aurita." eLife 8 (December). https://doi.org/10.7554/elife.50084.Romanes, G. J. 1885. Jelly-Fish, Star-Fish and Sea-Urchins: Being a Research on Primitive Nervous Systems. Appleton.Wang, Hengji, Joshua Swore, Shashank Sharma, John R. Szymanski, Rafael Yuste, Thomas L. Daniel, Michael Regnier, Martha M. Bosma, and Adrienne L. Fairhall. 2023. "A Complete Biomechanical Model of hydra Contractile Behaviors, from Neural Drive to Muscle to Movement." Proceedings of the National Academy of Sciences 120 (March). https://doi.org/10.1073/pnas.2210439120.Weir, Keiko, Christophe Dupre, Lena van Giesen, Amy S-Y Lee, and Nicholas W Bellono. 2020. "A Molecular Filter for the Cnidarian Stinging Response." eLife 9 (May). https://doi.org/10.7554/elife.57578.

We appreciate the excellent suggestion to further discuss non-neuronal adaptations in the context of studying the evolution of behavior. We have added additional text to the Discussion to cover this interesting field.